# A high-resolution analysis of arrestin2 interactions responsible for CCR5 endocytosis

Ivana Petrovic*, Samit Desai, Polina Isaikina[†], Layara Akemi Abiko, Anne Spang*, Stephan Grzesiek*

University of Basel, Basel, Switzerland

## eLife Assessment

The authors investigate arrestin2-mediated CCR5 endocytosis in the context of clathrin and AP2 contributions. Using an extensive set of NMR experiments, and supported by microscopy and other biophysical assays, the authors provide **compelling** data on the roles of AP2 and clathrin in CCR5 endocytosis. This **important** work will appeal to an audience beyond those studying chemokine receptors, including those studying GPCR regulation and trafficking. The distinct role of AP2 and not clathrin will be of particular interest to those studying GPCR internalization mechanisms.

*For correspondence:
ivana.petrovic@unibas.ch (IP);
anne.spang@unibas.ch (AS);
stephan.grzesiek@unibas.ch (SG)

Present address: [†]Center for Life Sciences, Paul Scherrer Institut, Villigen, Switzerland

Competing interest: The authors declare that no competing interests exist.

## Abstract

Clathrin-mediated endocytosis (CME) is crucial for regulating G protein-coupled receptors (GPCRs) via phosphorylation-dependent arrestin interactions. Despite detailed structural knowledge on the arrestin interactions with phosphorylated tails of GPCRs, the interplay between receptor phosphorylation and arrestin coupling to the CME machinery is not well understood, in particular due to the weakness and dynamics of the individual molecular interactions. Here, we have characterized the interactions of arrestin2, which is activated by the phosphorylated C-terminus of the human chemokine receptor 5 (CCR5), with the main protein constituents of CME, namely clathrin and AP2, by solution NMR spectroscopy, biochemical, and cellular assays. The NMR analysis revealed that arrestin2 interacts weakly with clathrin through a single binding site, independent of arrestin2 activation. In contrast, the arrestin2-AP2 interaction is stronger, requires arrestin2 activation by the CCR5 phospho-tail, and depends quantitatively on its degree of phosphorylation. These in vitro results are corroborated by cellular assays, which show that the chemokine-induced formation of a long-lived CCR5-arrestin2 internalization complex depends strongly on the interaction of arrestin2 with AP2, but not with clathrin. Taken together, these findings provide quantitative, atom-scale insights on the first steps of CCR5 endocytosis.

## Introduction

G protein-coupled receptors (GPCRs) represent a large family of transmembrane cell surface receptors that mediate the cellular response to a wide variety of signals. Their crucial role in many human physiological processes makes them an important drug target with about a third of existing small-molecule drugs binding to GPCRs and an enormous potential for further development (*Congreve et al., 2020*). To ensure precise signal transduction, the receptor signaling cascade events must be tightly regulated. This is achieved by coordinating receptor interactions with various intracellular proteins, including G proteins (*Weis and Kobilka, 2018*), GPCR kinases (GRKs) (*Komolov and Benovic, 2018*), and arrestins (*Gurevich and Gurevich, 2019*), with the latter eventually leading to receptor internalization in a process known as endocytosis. Endocytosis is a major mechanism of GPCR signal downregulation

(*Hanyaloglu and von Zastrow, 2008*). However, some receptors rapidly recycle back to the plasma membrane (*Hanyaloglu and von Zastrow, 2008*), which may also serve to scavenge external ligands and degrade them within the lysosomes (*Gu et al., 2023*; *Otero et al., 2006*).

Although GPCR endocytosis can occur through several different pathways, the vast majority of GPCRs are internalized by clathrin-mediated endocytosis (CME) (*Wolfe and Trejo, 2007*). This process relies on the clathrin protein for the formation of vesicular structures from the plasma membrane that later mature into endosomes (*Kaksonen and Roux, 2018*; *Smith and Smith, 2022*). While clathrin exhibits a diverse interaction spectrum (*Dell'Angelica, 2001*; *Lemmon and Traub, 2012*), it cannot directly interact with the membrane receptors or plasma membrane lipids and requires adaptor proteins to select its internalization cargo (*Kovtun et al., 2020*; *Popova et al., 2013*; *Smith et al., 1998*; *Wolfe and Trejo, 2007*). The majority of GPCRs require arrestins as endocytic adaptors for internalization (*Goodman et al., 1996*). The interaction of GPCRs with arrestins is triggered by the phosphorylation of the GPCR C-terminal tail or intracellular loops by GRKs (*Gurevich and Gurevich, 2019*). Only two non-visual arrestins, arrestin2 and arrestin3, bind to hundreds of non-visual GPCRs (*Gurevich and Gurevich, 2019*). This is then followed by the interaction with the endocytic machinery (*Moo et al., 2021*), in particular clathrin (*Goodman et al., 1997*; *Krupnick et al., 1997*) and the adaptor complex 2 protein (AP2), which plays a central role in the formation of the clathrin-coated vesicles (*Kelly et al., 2014*) and the selection of the cargo molecules (*Schmid et al., 2006*).

We have recently investigated the phosphorylation and arrestin2 interactions of the human chemokine receptor 5 (CCR5) (*Isaikina et al., 2023*). CCR5 is a GPCR, which acts as the main HIV co-receptor (*Alkhatib, 2009*) and plays a central role in inflammation (*Zeng et al., 2022*), COVID-19 infection (*Patterson et al., 2021*), chronic diseases (*Zeng et al., 2022*), as well as cancer (*Jiao et al., 2019*). This revealed a key pXpp GPCR phosphorylation motif in the CCR5 C-terminal tail, which is specifically recognized by well-defined interactions of individual phosphates with positively charged arrestin2 residues. The pXpp motif is widely observed in other receptors (*Isaikina et al., 2023*; *Maharana et al., 2023*) known to form a stable complex with arrestin2 (*Bous et al., 2022*; *Huang et al., 2020*; *Kang et al., 2015*). The strength of the arrestin interaction correlates with the phosphosite density and its distribution on the receptor C-terminal tail and intracellular loops (*Isaikina et al., 2023*; *Oakley et al., 2001*).

While our previous work revealed the atomic details of the interactions between arrestin and multiple GPCR phosphosites leading to arrestin activation, the interplay between the activated arrestin with clathrin and AP2, which causes receptor internalization, remains poorly understood. Here, we have characterized the interactions of arrestin2 with these two proteins as a function of arrestin2 activation by the phosphorylated CCR5 C-terminal tail using NMR and biochemical assays. The results establish strong activation-dependent interactions of arrestin2 with AP2, but not with clathrin. These in vitro results are validated by cellular assays, which also demonstrate that CCR5 forms long-lived endocytosed complexes with arrestin2 upon agonist chemokine stimulation.

## Results

### Localizing and quantifying the arrestin2-clathrin interaction by NMR spectroscopy

The CCR5 receptor relies on arrestin2/3 for internalization (*Fraile-Ramos et al., 2003*) via clathrin-mediated pathways (*Signoret et al., 2005*). Arrestin2 consists of two consecutive (N- and C-terminal) β-sandwich domains (*Figure 1A*), followed by the disordered clathrin-binding loop (CBL, residues 353–386), strand β20 (residues 386–390), and a disordered C-terminal tail after residue 393. The CBL contains a clathrin-binding motif ([L/I]ΦXΦ[D/E]) where X denotes any amino acid and Φ is a bulky hydrophobic residue (*Dell'Angelica, 2001*; *Smith et al., 2017*; *Figure 1A*, residues 376–380, *Figure 1—figure supplement 1A*). Strand β20 forms a characteristic β-sheet with the N-terminal strand β1 in inactive arrestin. Upon activation, the phosphorylated receptor tail replaces β20 in the β-sheet and releases it from the arrestin core.

Clathrin and arrestin interact in their basal state (*Goodman et al., 1996*), and a structure of a complex between arrestin2 and the clathrin heavy chain N-terminal domain (residues 1–363, named clathrin-N in the following) has been solved by X-ray crystallography in the absence of an arrestin2-activating phosphopeptide (PDB: 3GD1) (*Kang et al., 2009*). This structure (*Figure 1—figure*

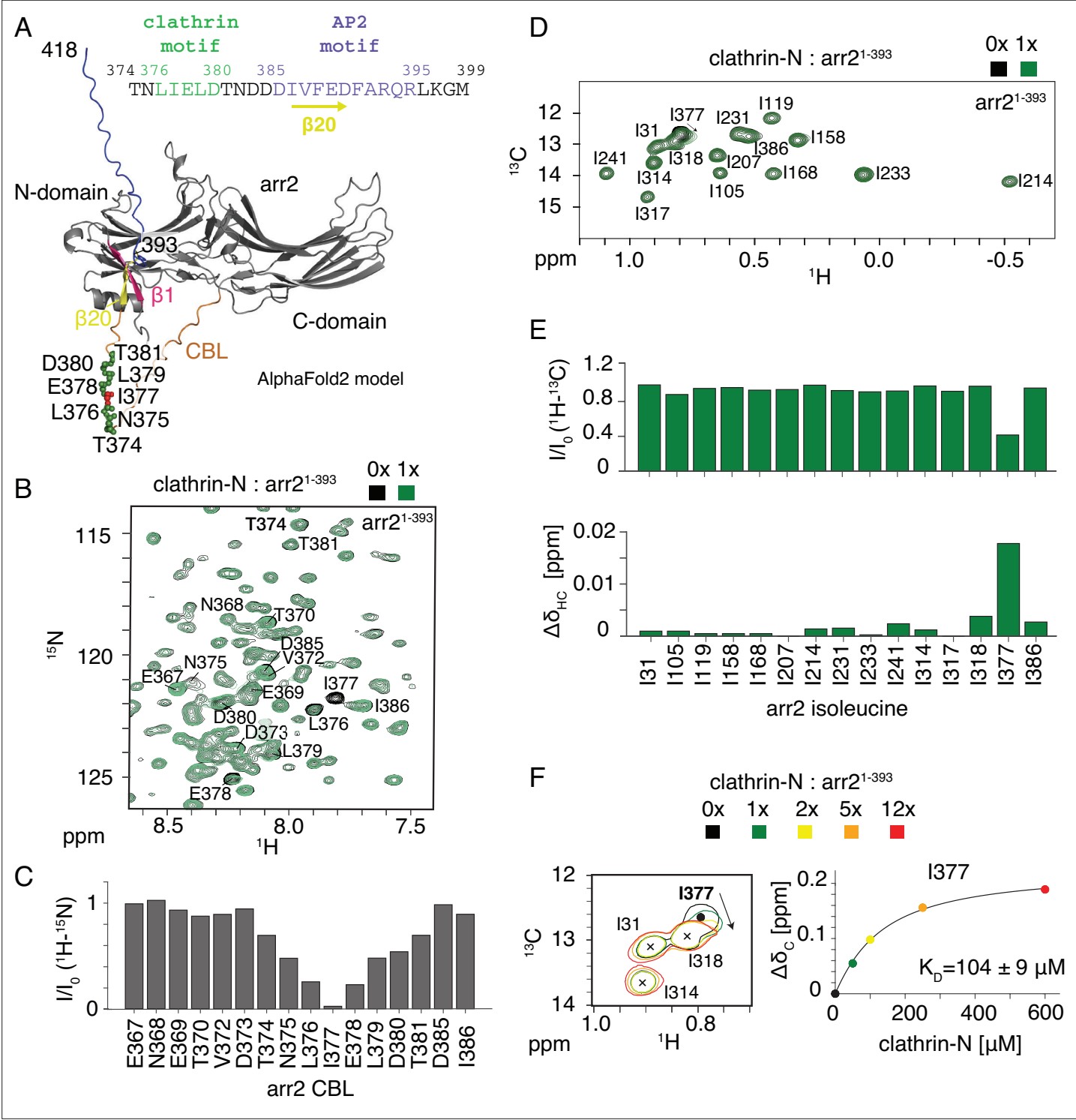

**Figure 1.** Arrestin2-clathrin-N interactions detected by arrestin2 NMR spectra. (**A**) AlphaFold2 model of arrestin2[1-418] (arr2) showing the N- and C-domain (gray) followed by the clathrin-binding loop (CBL, orange), β20 strand (yellow), and the disordered C-terminal tail (blue). The model was used to visualize the clathrin-binding loop and the 344-loop of the arrestin2 C-domain, which are not detected in the available crystal structures of apo arrestin2 [bovine: PDB 1G4M (*Han et al., 2001*), human: PDB 8AS4 (*Isaikina et al., 2023*)] due to flexibility. In the other structured regions, the model is virtually identical to the crystal structures. The C-terminal strand β20 forms a parallel β-sheet with the N-terminal strand β1 (pink) in inactive arrestin2 and is released from arrestin2 upon activation by phosphopeptides. Arrestin2 residues interacting with clathrin-N are shown as green (red for main interacting residue I377) spheres. The clathrin and AP2 binding motifs are indicated in the arrestin2 sequence on the top. (**B**) Part of ${}^{1}$H-${}^{15}$N TROSY spectrum of apo arrestin2[1-393] (black) and upon addition of an equimolar amount of clathrin-N (green). Assigned CBL arrestin2[1-393] residues are indicated. (**C**) Intensity ratios of

*Figure 1 continued on next page*

*Figure 1 continued*

assigned complexed vs. apo arrestin2[1-393] CBL $^1$H-$^{15}$N resonances. The region between T374 and T381 undergoes significant intensity attenuation upon clathrin-N binding. (**D**) $^1$H-$^{13}$C HMQC spectra of apo Ile-δ1-$^{13}$CH$_3$, $^2$H-labeled arrestin2[1-393] (black) and upon addition of an equimolar amount of clathrin-N (green). (**E**) Intensity attenuation (top) and chemical shift perturbation (bottom) of arrestin2[1-393] isoleucine $^1$H$_3$-$^{13}$C$^{δ1}$ resonances upon clathrin-N addition. Out of 15 isoleucine residues, significant changes are observed only for I377. (**F**) Left: part of $^1$H-$^{13}$C HMQC spectrum showing resonance shifts of I377 upon clathrin-N binding. Right: detected chemical shift changes as a function of clathrin-N concentration. The solid line depicts a non-linear least-squares fit to the data points with respective dissociation constant.

The online version of this article includes the following source data and figure supplement(s) for figure 1:

**Source data 1.** Values of the intensity ratios of assigned complexed vs. apo arrestin2[1-393] clathrin-binding loop (CBL) resonances for *Figure 1C*, intensity ratios and chemical shift perturbation of arrestin2[1-393] isoleucine $^1$H$_3$-$^{13}$C$^{δ1}$ resonances upon clathrin-N addition for *Figure 1E*, and detected chemical shift changes as a function of clathrin-N concentration for *Figure 1F*.

**Figure supplement 1.** Arrestin sequence alignment and analysis of the arrestin2-clathrin-N complex structure.

**Figure supplement 2.** Mapping of clathrin-N interaction on arrestin2.

**Figure supplement 2—source data 1.** Values of the intensity ratios of complexed vs. apo arrestin2[1-393] resonances upon addition of clathrin N for panel B.

---

supplement 1B*) suggests a 2:1 binding model between arrestin2 and clathrin-N. The first interaction (site I) is observed between the $^{376}$LIELD$^{380}$ clathrin-binding motif of the arrestin2 CBL and the edge of the first two β-sheet blades of clathrin-N, whereas the second interaction (site II) occurs between arrestin2 residues $^{334}$LLGDLA$^{339}$ and the fourth and fifth blade of clathrin-N. The latter arrestin interaction site is not present in the arrestin2 splice variant arrestin2S (for short) where an 8-amino acid insert (residues 334–341) between β-strands 18 and 19 is removed (*Kang et al., 2009*).

To elucidate the clathrin-arrestin interactions under solution conditions and the influence of arrestin activation, we utilized NMR spectroscopy. *Figure 1B* depicts part of a $^1$H-$^{15}$N TROSY spectrum (full spectrum in *Figure 1—figure supplement 2A*) of the truncated $^{15}$N-labeled arrestin2 construct arrestin2[1-393] (residues 1–393), which encompasses the C-terminal strand β20, but lacks the disordered C-terminal tail. Due to intrinsic microsecond dynamics, the assignment of the arrestin2[1-393] $^1$H-$^{15}$N resonances by triple resonance methods is largely incomplete, but 16 residues (residues 367–381, 385–386) within the mobile CBL could be assigned. This region of arrestin is typically not visible in either crystal or cryo-EM structures due to its high flexibility. The addition of an equimolar amount of clathrin-N (clathrin heavy chain 1, residues 1–364) to arrestin2[1-393] induced significant line broadening from residue T374 to T381, with I377 being most strongly affected (*Figure 1C*, *Figure 1—figure supplement 2B*). This region almost exactly matches and fully contains the canonical arrestin2 clathrin-binding motif $^{376}$LIELD$^{380}$. *Figure 1A* shows these interacting residues mapped on an arrestin2 AlphaFold2 model (*Jumper et al., 2021*).

As $^1$H-$^{13}$C methyl resonances have more favorable relaxation properties than backbone $^1$H-$^{15}$N resonances (*Tugarinov et al., 2003*), we used Ile-δ1-$^{13}$CH$_3$-labeled arrestin2 to quantitatively monitor the formation of the arrestin2-clathrin complex (arrestin2 assignments transferred from BMRB, ID:51131; *Shiraishi et al., 2021*). HMQC spectra of the 15 Ile δ1-$^{13}$CH$_3$ methyl groups of arrestin2[1-393] (*Figure 1D*) revealed changes in peak intensity (*Figure 1E*, top) and chemical shift perturbations (*Figure 1E*, bottom) only for I377 at an equimolar addition of unlabeled clathrin-N to arrestin2[1-393]. An apparent dissociation constant $K_D$ = 104 ± 9 µM was determined from titrating clathrin-N and fitting the arrestin2 I377 $^{13}$C chemical shifts with a 1:1 binding isotherm (*Figure 1F*). Although the formed complex (total molecular weight 85 kDa) has low affinity, such interactions are common in endocytosis, which involves many multi-site interactions within a large network of proteins (*Smith et al., 2017*).

## The arrestin2-clathrin interaction is independent of arrestin2 activation

Due to significant line broadening and peak overlap of the arrestin2 resonances upon phosphopeptide addition, the influence of arrestin activation on the clathrin interaction could not be detected on either backbone or methyl resonances. However, clathrin-N, although similar in size to arrestin2[1-393] (41 kDa vs. 44 kDa), provided considerably better $^1$H-$^{15}$N TROSY spectra due to the absence of significant micro- to millisecond dynamics. This allowed us to observe the effect of arrestin2 activation on its interaction with clathrin by titrating arrestin2[1-393] to $^{15}$N-labeled clathrin-N.

*Figure 2A* (left) shows the intensity changes (full spectra in *Figure 2—figure supplement 1A*) of the clathrin-N $^1$H-$^{15}$N TROSY resonances [assignments transferred from BMRB, ID:25403, *Zhuo et al.,*

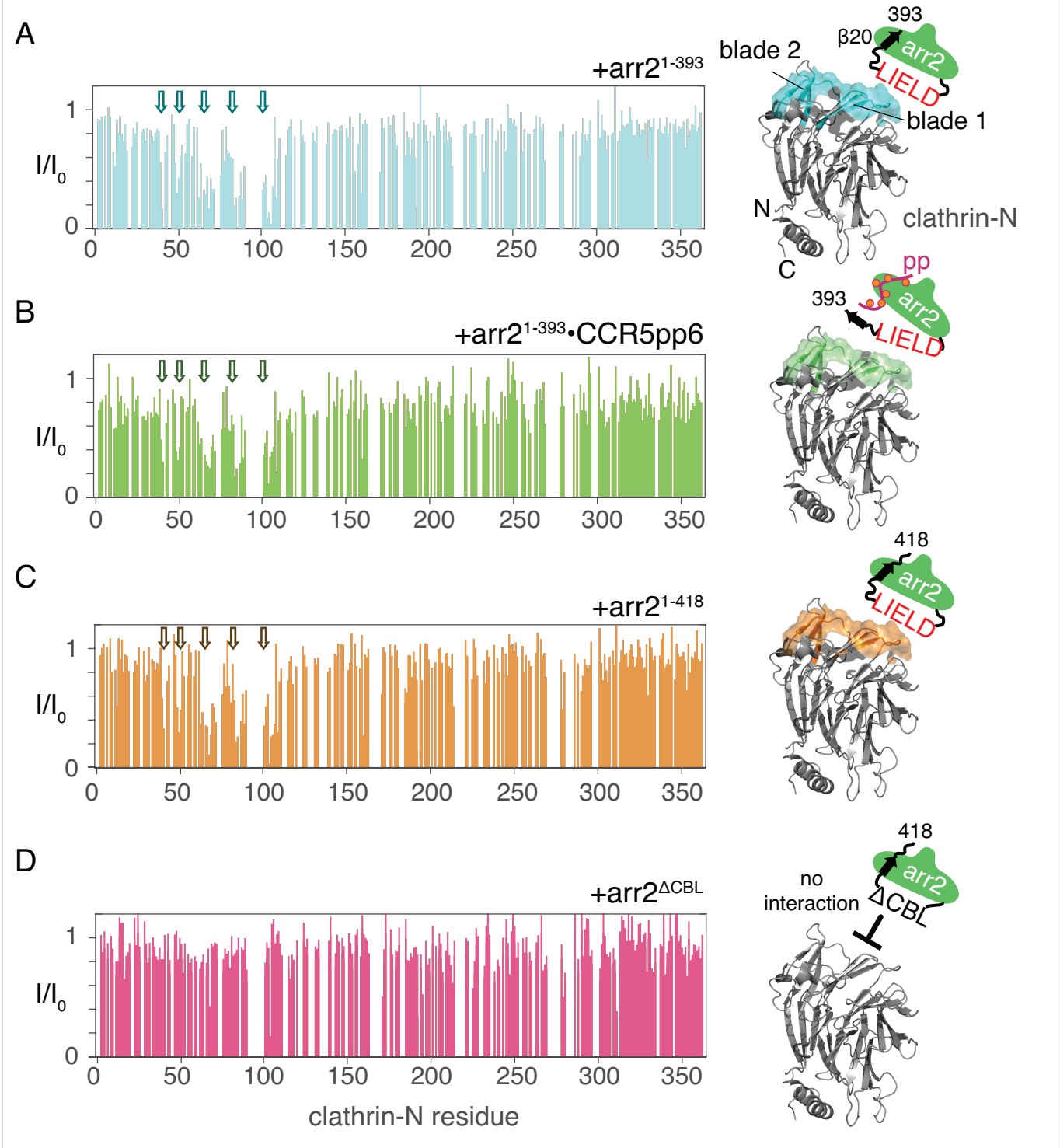

**Figure 2.** Arrestin2-clathrin-N interactions detected by clathrin-N NMR spectra. (Left): Intensity attenuation $I/I_0$ of clathrin-N $^1$H-$^{15}$N TROSY resonances upon addition of equimolar amounts of various arrestin2 constructs: (**A**) apo arrestin2$^{1-393}$, (**B**) CCR5pp6-activated arrestin2$^{1-393}$, (**C**) full-length arrestin2$^{1-418}$, and (**D**) arrestin2$^{\Delta CBL}$ with removed clathrin-binding motif. Regions that undergo significant attenuation upon interaction with arrestin2 are indicated by arrows. Right: Residues undergoing significant intensity attenuation (one standard deviation below the average signal attenuation) are marked on the structure of clathrin-N (PDB: 3GD1) together with a schematic representation of the respective arrestin2 construct.

The online version of this article includes the following source data and figure supplement(s) for figure 2:

**Source data 1.** Values of the intensity ratios of complexed vs. apo clathrin N resonances upon addition of various arrestin2 constructs for *Figure 2A–D*.

**Figure supplement 1.** Mapping of arrestin2 interaction on clathrin-N.

2015] upon addition of a one-molar equivalent of arrestin2$^{1-393}$. A significant intensity reduction due to line broadening is detected for clathrin-N residues 39–40, 48–50, 62–72, 83–90, 101–106, and 108. These residues form a clearly defined binding region at the edges of blade 1 and blade 2 of clathrin-N (*Figure 2A*, right), which corresponds to interaction site I in the 3GD1 crystal structure, involving the conserved arrestin2 $^{376}$LIELD$^{380}$ motif. However, no significant signal attenuation was observed for clathrin-N residues in blade 4 and blade 5, which would correspond to the crystal interaction site II with arrestin2 residues $^{334}$LLGDLA$^{339}$ that are absent in the arrestin2S splice variant. Thus, only one arrestin2 binding site in clathrin-N is detected in solution, and site II of the crystal structure may be a result of crystal packing.

Next, we tested whether the arrestin2-clathrin-N interaction is enhanced in the presence of the fully phosphorylated CCR5 phosphopeptide (CCR5pp6, where 6 indicates the number of phospho-sites, *Supplementary file 1*), which displaces arrestin2 strand β20 by forming an antiparallel β-sheet with strand β1, thereby activating arrestin2 (*Isaikina et al., 2023*). However, no additional signal atten-uation of the clathrin-N $^1$H-$^{15}$N resonances or extension to further clathrin-N regions was observed (*Figure 2B*, *Figure 2—figure supplement 1B*), when arrestin2$^{1-393}$ was added in the presence of a 5-molar equivalent of CCR5pp6 (corresponding to 80% of arrestin activation). Thus, activation of arrestin2 has apparently no effect on its interaction with clathrin-N. These results suggest that even for receptors with low phosphosite density, such as adrenergic and dopamine receptors (*Isaikina et al., 2023*; *Oakley et al., 2001*), the arrestin2-clathrin interaction may still take place, allowing efficient receptor internalization.

Previous studies have suggested that the last residues of the arrestin2 C-terminus have an inhibi-tory effect on clathrin binding (*Kern et al., 2009*). We, therefore, tested the interaction of $^{15}$N-labeled clathrin-N with full-length arrestin2 (arrestin2$^{1-418}$). However, the obtained residue intensity attenuation was indistinguishable from the one observed for the arrestin2$^{1-393}$ construct (*Figure 2C*, *Figure 2—figure supplement 1C*). Therefore, the residues of the arrestin2 C-terminal tail do not interfere with clathrin binding. In contrast, the interaction of clathrin with arrestin2$^{1-418}$ (*Figure 2D*, *Figure 2—figure supplement 1D*) was completely abolished when the arrestin2 clathrin-binding loop was replaced by the loop of visual arrestin (arrestin2$^{\Delta CBL}$, construct details in Methods). The latter loop lacks the clathrin-binding motif, making visual arrestin incapable of interacting with clathrin (*Goodman et al., 1997*).

Taken together, the solution NMR experiments reveal that arrestin2 interacts with clathrin-N through a single binding site, independently of arrestin2 activation and the conformation of the arrestin2 core.

## Arrestin activation is essential for AP2 interaction

We then proceeded to characterize the interaction of arrestin2 with AP2. AP2 consists of two large (α and β2) and two smaller (μ2 and σ2) subunits (*Collins et al., 2002*). While each subunit plays a role in CME (*Smith et al., 2017*), direct recognition of endocytic cargo motifs has been attributed to the σ2 subunit (LL-motif) and μ2 subunit (YxxΦ-motif) (*Jackson et al., 2010*). The β2 subunit of the AP2 complex (AP2β2) interacts with the arrestin2/3 C-terminus (*Kim and Benovic, 2002*; *Laporte et al., 2002*; *Laporte et al., 2000*), and a crystal structure of the complex between the AP2β2 C-terminal domain and a C-terminal arrestin2 peptide (residues 383–402) encompassing strand β20 has been solved (PDB: 2IV8) (*Schmid et al., 2006*). Interestingly, the C-terminal arrestin2 peptide forms a helical structure in the complex with AP2 (*Figure 3A*) as opposed to its β-sheet coordination with strand β1 in inactive arrestin2. No experimental data exist on the formation of the AP2-arrestin2 complex in the context of full-length arrestin2 and its activation by phosphorylated receptor tails.

To test how different phosphorylation levels influence arrestin2 recruitment by the C-terminal domain of AP2β2 (AP2β2$^{701-937}$, residues 701–937), we performed a SEC binding assay. In the absence of phosphopeptide, an equimolar mixture of 10 μM arrestin2$^{1-418}$ and 10 μM AP2β2 is separated into two peaks (*Figure 3B*, orange), which elute at the positions corresponding to the individual apo proteins (*Figure 3B*, blue, green). Upon addition of 100 μM CCR5pp6 (*Figure 3C*, purple), an addi-tional SEC peak appears at an elution volume of 2.6 ml, which corresponds to the formed arrestin2-AP2β2 complex. Monotonously decreasing complex yields were observed when reducing the amounts of CCR5pp6 added to the mixture (*Figure 3D*), and no complex formation occurred in the presence of 100 μM non-phosphorylated CCR5 phosphopeptide (CCR5pp0) (*Figure 3E*). The dissociation constant of the CCR5pp6-arrestin2 complex was determined previously as 45 μM (*Supplementary*

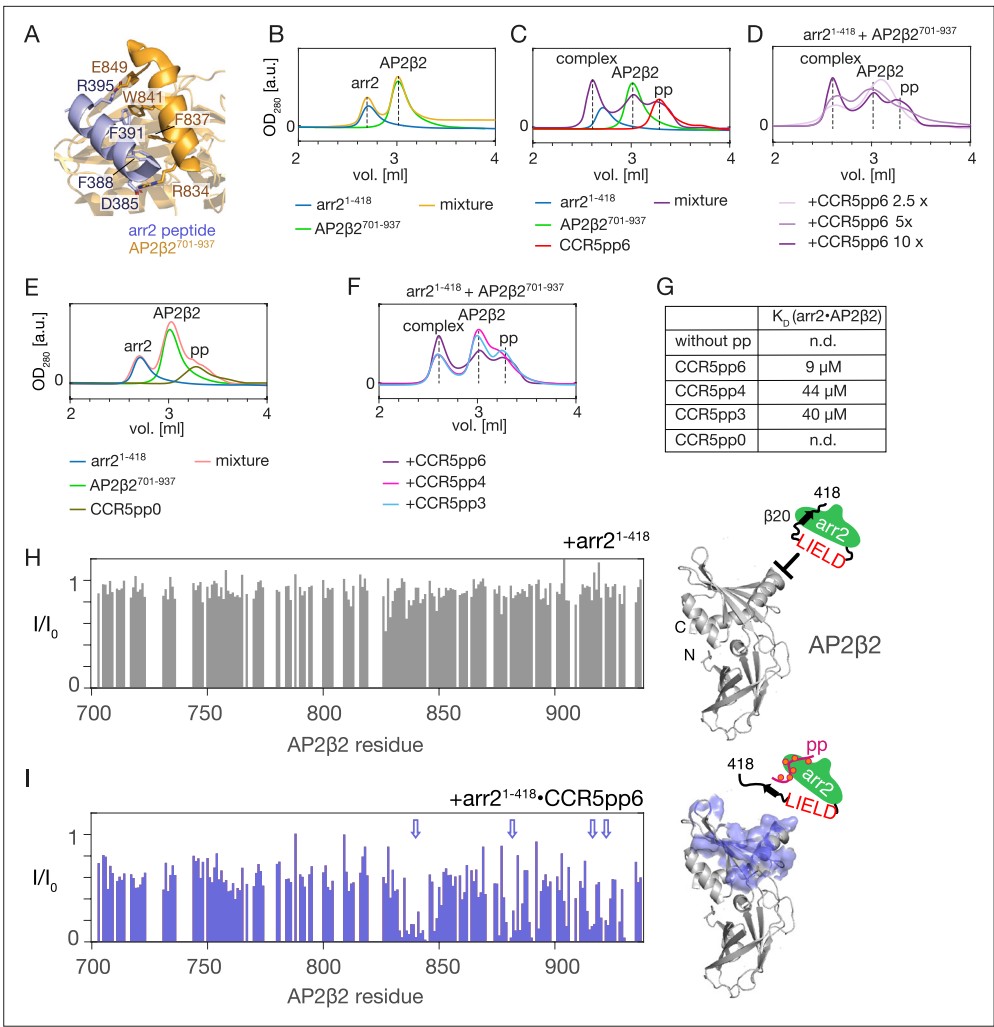

**Figure 3.** Arrestin2 interaction with the C-terminal domain of the AP2β2. (**A**) Structure of AP2β2$^{701-937}$ (orange) in complex with arrestin C-terminal peptide (blue) (PDB: 2IV8). Important residues from both chains stabilizing the interaction are depicted in stick representation. (**B–F**) SEC profiles of arrestin2$^{1-418}$, AP2β2$^{701-937}$, various phosphopeptides, and their mixtures. The annotated color coding below the SEC profile indicates the individual sample composition. In panels (**B, C, E**), 'mixture' refers to a SEC sample containing all of the individual components indicated on the left. In panels (**D, F**), the primary sample composition is indicated above the SEC profile, and the annotated color coding below the SEC profile indicates the added component to the primary sample composition. (**G**) Apparent affinities for arrestin2-AP2 complex formation in the presence of phosphopeptides derived by integrating the SEC peaks marked in *Figure 4F* and scaling the integrals by the respective extinction coefficients. 'nd' indicates 'not detectable.' (**H, I**) (Left): intensity attenuation $I/I_0$ of AP2β2$^{701-937}$ $^1$H-$^{15}$N TROSY resonances upon addition of equimolar amounts of apo arrestin2$^{1-418}$ (**H**) or CCR5pp6-activated arrestin2$^{1418}$ (**I**). Regions that undergo extensive specific attenuation upon interaction with arrestin2 are indicated by arrows. Right: Residues undergoing extensive specific intensity attenuation are marked on the structure of AP2β2 (PDB: 2IV8) together with a schematic representation of the respective arrestin2 construct.

The online version of this article includes the following source data and figure supplement(s) for figure 3:

**Source data 1.** Values of the intensity ratios of complexed vs. apo AP2β2 resonances upon addition of apo or peptide-bound arrestin2 for *Figure 3H and I*.

**Figure supplement 1.** Arrestin2 interaction with the C-terminal domain of AP2β2 detected by NMR.

**Figure supplement 2.** Arrestin2 interaction with clathrin-N and AP2β2 monitored by SEC.

**Figure supplement 2—source data 1.** PDF file containing original SDS-PAGE gel for *Figure 3—figure supplement 2* indicating the relevant bands.

**Figure supplement 2—source data 2.** Original file for SDS-PAGE gel analysis displayed in *Figure 3—figure*

*Figure 3 continued on next page*

*Figure 3 continued*

*supplement 2*.

**Figure supplement 3.** Arrestin2[1-418] trypsin proteolysis assay.

**Figure supplement 3—source data 1.** PDF file containing original SDS-PAGE gels for *Figure 3—figure supplement 3* indicating the relevant bands.

**Figure supplement 3—source data 2.** Original files for SDS-PAGE gel analysis displayed in *Figure 3—figure supplement 3*.

*file 1*; *Isaikina et al., 2023*). Not surprisingly, the formation of the arrestin2-AP2β2 complex depends quantitatively on the affinity of the phosphopeptide-arrestin2 complex. Correspondingly, the addition of the less phosphorylated CCR5pp3 and CCR5pp4 with lower affinities towards arrestin2 (both $K_D$ ~200 μM, *Supplementary file 1*; *Isaikina et al., 2023*) resulted in a considerable, similar reduction of the formed arrestin2-AP2β2 complex (*Figure 3F*).

Using the extinction coefficients of all involved polypeptides, the apparent affinities between AP2β2 and arrestin2 could be determined from the SEC profiles (*Figure 3G*, see Materials and methods for details). In the presence of 100 μM CCR5pp6, this apparent $K_D$ amounts to 9 μM. An identical affinity value for the affinity between AP2β2 and an isolated arrestin β20 peptide (residues 383–402, D402C) has been observed by EPR spectroscopy (*Moaven et al., 2013*). In the presence (100 μM) of the less phosphorylated CCR5 peptides, the apparent AP2β2-arrestin2 affinities are reduced by ~4–5 times (CCR5pp4: 44 μM, CCR5pp3: 40 μM), in good agreement with their ~4–5 times reduced affinity for arrestin2.

We also attempted to stabilize the arrestin2-AP2β2-phosphopeptide complex through the addition of PIP2, which can stabilize arrestin complexes with the receptor (*Janetzko et al., 2022*). The addition of PIP2 increased the formation of arrestin2 dimers and higher oligomers, presumably due to the presence of additional charges. Unfortunately, the resolution of the SEC experiments was not sufficient to separate the arrestin2 oligomers from complexes with AP2β2.

It is interesting to note that CCR5pp3 and CCR5pp4 have similar affinities towards arrestin2 despite their different phosphorylation levels (threefold vs. fourfold phosphorylated), but only CCR5pp3 contains the pXpp barcode which increases the arrestin2 activation, as assayed by the binding of the Fab30 antibody (*Isaikina et al., 2023*). Apparently, this barcode does not play a significant role in the formation of the arrestin2-AP2β2 complex, as the complex formation appears solely governed by the affinity of the phosphopeptide towards arrestin2.

To observe the atomic details of arrestin2 activation on its interaction with AP2β2, we used [15]N-labeled AP2β2[701-937] [assignment transferred from BMRB ID: 52320; *Naudi-Fabra et al., 2024*]. Upon addition of arrestin2[1-418], no changes in the AP2β2 [1]H-[15]N spectra are detected (*Figure 3H*, *Figure 3—figure supplement 1A*), apparently due to the unavailability of arrestin2 β20 strand and C-terminal tail. However, when arrestin2[1-418] was added in the presence of a 5-molar equivalent of CCR5pp6 (corresponding to 80% of arrestin activation), a significant intensity reduction due to line broadening is detected for many AP2β2 [1]H-[15]N resonances (*Figure 3I*, left, *Figure 3—figure supplement 1B*). The strongest attenuated resonances clearly define a binding region within the C-terminal domain of AP2β2 (*Figure 3I*, right), which is an extension of the binding surface observed in the crystal structure (PDB: 2IV8) of AP2β2 in complex with a 21-residue arrestin2 C-terminal peptide containing strand β20 (*Schmid et al., 2006*). In agreement with the AP2β2 NMR observations, no interaction was observed in the arrestin2 methyl and backbone NMR spectra upon addition of AP2β2 in the absence of phosphopeptide (*Figure 3—figure supplement 1C and D*). However, the significant line broadening of the arrestin2 resonances upon phosphopeptide addition (*Figure 3—figure supplement 1E and F*) precluded a meaningful assessment of the effect of the AP2β2 addition on arrestin2 in the presence of phosphopeptide. The observed line broadening of arrestin2 in the presence of phosphopeptide must be a result of increased protein motions and is not caused by a decrease in protein stability, since the melting temperature of arrestin2 in the absence and presence of phosphopeptide are identical (56.9 ± 0.1°C).

We then tested whether the presence of clathrin-N influences complex formation between arrestin2 and AP2β2, and vice versa. Arrestin2[1-418] does not form a complex in the SEC assay with clathrin-N (*Figure 3—figure supplement 2A*), which agrees with the weak affinity observed in the

NMR experiments (*Figure 1F*). Furthermore, a SEC analysis of a mixture of arrestin2[1-418], clathrin-N, AP2β2[701-937], and CCR5pp6 (*Figure 3—figure supplement 2B*) indicates formation of a complex at 2.6 ml elution volume. An SDS analysis of this complex fraction showed the presence of arrestin2[1-418] and AP2β2[701-937], but not of clathrin-N (*Figure 3—figure supplement 2C*). This indicates that clathrin-N does not interact strongly with the arrestin2-AP2-CCR5 tail complex. As such, clathrin may not engage with the initial receptor internalization complex, but may be recruited at later stages of the CME.

To test the stability of the formed arrestin2-AP2 interaction and its quaternary coordination, we performed a trypsin proteolysis assay (*Figure 3—figure supplement 3*, uncropped gels in *Figure 3—figure supplement 3—source data 1* and *Figure 3—figure supplement 3—source data 2*). Trypsin recognizes arrestin2 residue R393 (*Xiao et al., 2004*), which is buried in the arrestin2 apo state, but is involved in the interaction with AP2β (*Laporte et al., 2000*). Without the phosphopeptide, the arrestin2[1-418] digestion midpoint occurs at 20–30 min due to the inaccessibility of the cleavage site (*Figure 3—figure supplement 3A–C*). In the presence of a 5-molar equivalent of CCR5pp6 (*Figure 3—figure supplement 3A*), the top band corresponding to full-length arrestin2 has almost completely disappeared (relative abundance 19%) already 5 min after incubation as a consequence of the β20 strand release and the exposure of R393 to trypsin. However, the presence of AP2β2[701-937] protects arrestin[1-418] from proteolysis, as evident from significantly higher abundance at 5 min and a shift of the proteolysis midpoints to 10, 20, and 30 min at 1-, 2-, and 5-molar equivalents of AP2β2[701-937], respectively.

Analogous digestion experiments using CCR5pp3 (*Figure 3—figure supplement 3B*) or CCR5pp4 (*Figure 3—figure supplement 3C*) phosphopeptides revealed that they did not enhance the rate of arrestin2 proteolysis by trypsin at a 3-molar equivalent, as expected from their weaker affinity towards arrestin2. In agreement with this, however, upon addition of 25-molar equivalents of either peptide, the rate of arrestin2 proteolysis was significantly increased, as full-length arrestin2 almost completely disappeared already 5 min after incubation. This increase was abolished when adding AP2β2[701-937]. Thus, as for CCR5pp6, the presence of AP2β2[701-937] also protects arrestin2[1-418] activated by CCR5pp3 or CCR5pp4 from proteolysis.

In summary, these results indicate that arrestin2 activation by the binding of receptor phosphopeptides and the subsequent release of its strand β20 is required for the interaction with AP2. This interaction is modulated by the phosphorylation level of the receptor C-terminal peptide and its relative abundance. The formed AP2 complex with arrestin is stable and more resistant to proteolysis.

## Ligand-dependent CCR5 internalization monitored in HeLa cells

We then validated the obtained in vitro results on the CCR5-arrestin2-clathrin and -AP2 interactions by cellular assays. According to the strength and persistence of arrestin interactions, GPCRs have been categorized into class A and class B subtypes (*Isaikina et al., 2023*; *Oakley et al., 2000*). To distinguish these arrestin classes from the common IUPHAR classification of GPCRs, we refer in the following to them as arr-class A and B, respectively. Arr-class A receptors (e.g. adrenergic receptors) form transient complexes with arrestins that quickly dissociate near the plasma membrane, and the receptor traffics alone into endosomes. In contrast, arr-class B receptors (e.g. the vasopressin receptor 2, V2R) have a higher affinity towards arrestin and form long-lived complexes that traffic together to endosomes (*Oakley et al., 2000*). We have recently established that the phosphorylated CCR5 C-terminus forms a high-affinity complex with arrestin (*Isaikina et al., 2023*), thus assigning the CCR5 to the arr-class B subtype.

To better understand the stability of the CCR5-arrestin2 complex, we first monitored CCR5 internalization upon chemokine stimulation in fixed HeLa (CCL2, ATCC nomenclature) cells (*Figure 4*). HeLa cells were transiently transfected with plasmids containing C-terminally FLAG-tagged CCR5 and YFP-tagged arrestin2 (arrestin2-YFP) genes. This receptor construct was used to minimize CCR5 modifications and monitor the arrestin2 interaction in conditions as close to native as possible. Before the addition of the chemokine ligands (0 min), CCR5 was observed at the plasma membrane (magenta), while arrestin2 (green) was distributed throughout the cell (*Figure 4A–C*). Cells stimulated with the antagonist chemokine [5P12]CCL5 (*Gaertner et al., 2008*) retained this CCR5 and arrestin2 distribution even 30 min after ligand addition (*Figure 4A*). In contrast, when cells were stimulated with the native agonist chemokine CCL5 (*Hughes and Nibbs, 2018*; *Figure 4B*) or the super-agonist [6P4]

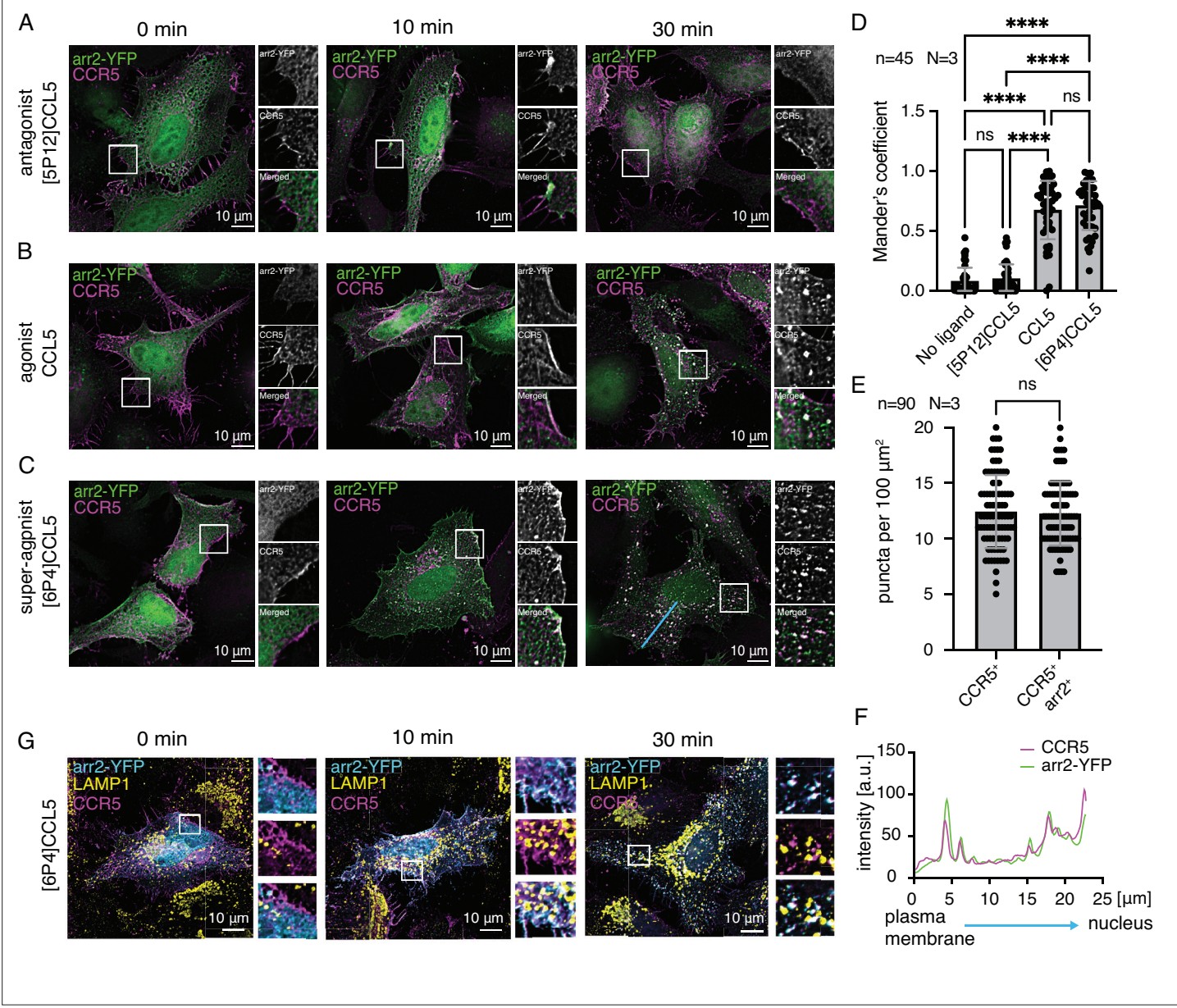

**Figure 4.** CCR5 internalization in the presence of the chemokine ligands. CCR5 internalization monitored in HeLa cells co-transfected with plasmids containing arrestin2-YFP (green) and CCR5 genes (magenta) stimulated with (**A**) [5P12]CCL5 (antagonist), (**B**) CCL5 (natural agonist), (**C**) [6P4]CCL5 (super-agonist). Cells were stimulated for 0, 10, and 30 min with chemokine ligands before fixation and preparation for fluorescence microscopy. Each image is accompanied by a zoomed region of interest (white squares) showing deconvolved CCR5 and arrestin2-YFP signals. (**D**) Mander's colocalization coefficients of the CCR5 and arrestin2-YFP signals in the absence and presence of chemokine ligands 30 min after stimulation. Individual values, as well as mean and standard deviation are shown for N=3 biological replicates and n=45 ROIs from 20 cells; ****$p<0.0001$; ns: not significant ($p>0.9999$). (**E**) Density of CCR5-positive (CCR5+) and CCR5/arrestin2-positive (CCR5+/arrestin2+) puncta after 30 min [6P4]CCL5 ligand stimulation. Individual values, as well as mean and standard deviation are shown for N=3 biological replicates and n=90 ROIs from 30 cells; ns: not significant ($p>0.9999$). (**F**) CCR5 and arrestin2 fluorescence signals along the trajectory from the plasma membrane to the nucleus (blue line in panel **c**) 30 min after [6P4]CCL5 ligand stimulation. (**G**) Lysosomal trafficking of the CCR5 (magenta) and arrestin2 (cyan) complex in the presence of [6P4]CCL5 monitored in the HeLa cells using LAMP1 antibodies (yellow). No recruitment to the lysosome is observed.

The online version of this article includes the following source data and figure supplement(s) for figure 4:

**Source data 1.** Values of Mander's colocalization coefficient, CCR5+ and CCR5+/arr2+ puncta counting, and individual CCR5 and arrestin2-YFP signal intensities for *Figure 4D–F*.

**Figure supplement 1.** Arrestin2 internalization (control) and re-localization in live HeLa cells upon chemokine stimulation in the presence of CCR5.

CCL5 (*Gaertner et al., 2008*; *Isaikina et al., 2021*; *Figure 4C*), recruitment of arrestin2 from the cytoplasm to the plasma membrane could be detected within 10 min. Prolonged ligand incubation times (30 min) lead to CCR5 internalization in complex with arrestin2. To eliminate the possibility that arrestin2 recruitment to the membrane was due to interactions with endogenous proteins in HeLa cells, we monitored internalization in live cells transfected only with arrestin2. In this case, the cells did not show characteristic arrestin2 recruitment to the plasma membrane or robust puncta formation, indicative of endosomes, even in the presence of the super-agonist (*Figure 4—figure supplement 1A*). This control experiment also revealed a few internal arrestin puncta devoid of receptor, presumably due to the arrestin2 overexpression. This phenomenon was also observed previously (*Beautrait et al., 2017*; *Oakley et al., 2000*).

Stronger phosphorylation and arrestin2/3 recruitment potencies have been reported for [6P4]CCL5 as compared to wild-type CCL5 (*Martins et al., 2020*). In agreement with this, [6P4]CCL5 appears more potent than CCL5 in recruiting arrestin2 to the plasma membrane and inducing CCR5-arrestin2 puncta formation after 10 min (*Figure 4B and C*). However, 30 min after stimulation, no significant difference in CCR5 and arrestin2 colocalization was observed as quantified by the Mander's coefficient (*Figure 4D*). The Mander's coefficient is about 0.7 in cells stimulated by either agonist, but close to zero for non-stimulated or antagonist-incubated cells (*Figure 4D*). Apparently, the two agonist ligands induce a differing initial rate of arrestin2 recruitment by CCR5. However, once arrestin2 is recruited, the CCR5-arrestin2 complex accumulates and remains stable for at least 30 min inside the cells.

To further corroborate the arr-class B-like behavior of CCR5, we counted the puncta positive for CCR5 or both CCR5 and arrestin2 in cells at this time point after stimulation by [6P4]CCL5 (*Figure 4E*). No significant difference was observed between the two values. This indicates that the number of puncta containing only CCR5, but not arrestin2, must be very small and that the majority of CCR5 puncta also contain arrestin2, consistent with stable complex formation. Furthermore, the CCR5 and arrestin2 colocalization occurs not only close to the plasma membrane but also close to the nucleus (*Figure 4F*), indicative of the arr-class B-like behavior of CCR5. For arr-class A receptors, it would be expected that arrestin2 is excluded from receptor-containing structures, which traffic into the perinuclear region (*Oakley et al., 2000*).

As both recycling (*Gu et al., 2023*; *Mueller et al., 2002*) and degradation (*Gu et al., 2023*) has been observed during CCR5 internalization, the cellular fate of the CCR5-arrestin2 complex was probed using the lysosomal-associated membrane protein 1 (LAMP1) as a marker of the lysosomal membranes (*Figure 4G*). No recruitment of the complex to the lysosomes (LAMP1, yellow) was detected in the presence of [6P4]CCL5 even 30 min after ligand stimulation. Similar results were obtained for both CCL5 and [6P4]CCL5 in live cells using the LysoTracker dye that accumulates in acidic environments, such as the lysosome (*Figure 4—figure supplement 1B and C*). These results indicate that the CCR5 internalization pathway, although mediated by arrestin, does not lead to rapid receptor degradation in lysosomes.

## Arrestin interaction with AP2 is essential for CCR5 internalization

The interactions of arrestin with clathrin and AP2 have mostly been studied for the internalization of the β2-adrenergic (β2AR) (*Barsi-Rhyne et al., 2022*; *Beautrait et al., 2017*; *Goodman et al., 1996*; *Kang et al., 2009*; *Kim and Benovic, 2002*) and V2R (*Barsi-Rhyne et al., 2022*; *Beautrait et al., 2017*) receptors. Here, we investigated how both interactions affect the internalization of CCR5 in a HeLa KO cell line (HeLa$^{-arr2/3}$) devoid of arrestin2 and arrestin3 as well as in HeLa (CCL2) cells (*Figure 5*, *Figure 5—figure supplement 1*, respectively).

To characterize the arrestin2-clathrin interaction, we generated a YFP-tagged arrestin2 construct with the clathrin-binding motif (arrestin2$^{\Delta LIELD}$-YFP) removed. No significant difference in CCR5 internalization was detected in either HeLa cell line between wild-type arrestin2-YFP and arrestin2$^{\Delta LIELD}$-YFP, when the cells were stimulated with [6P4]CCL5 (HeLa$^{-arr2/3}$: *Figure 5A–C*, HeLa: *Figure 5—figure supplement 1A*). Apparently, the arrestin2-clathrin interaction does not play a significant role in the internalization of the arr-class B CCR5 receptor. This is in contrast to findings for the arr-class A receptor β2AR where removal of the LIELD motif impairs receptor internalization (*Kim and Benovic, 2002*).

To test the influence of the arrestin2-AP2 interaction, we introduced mutations within the arrestin2 adaptin-binding motif, which is conserved across all arrestins (*Figure 1—figure supplement 1A*).

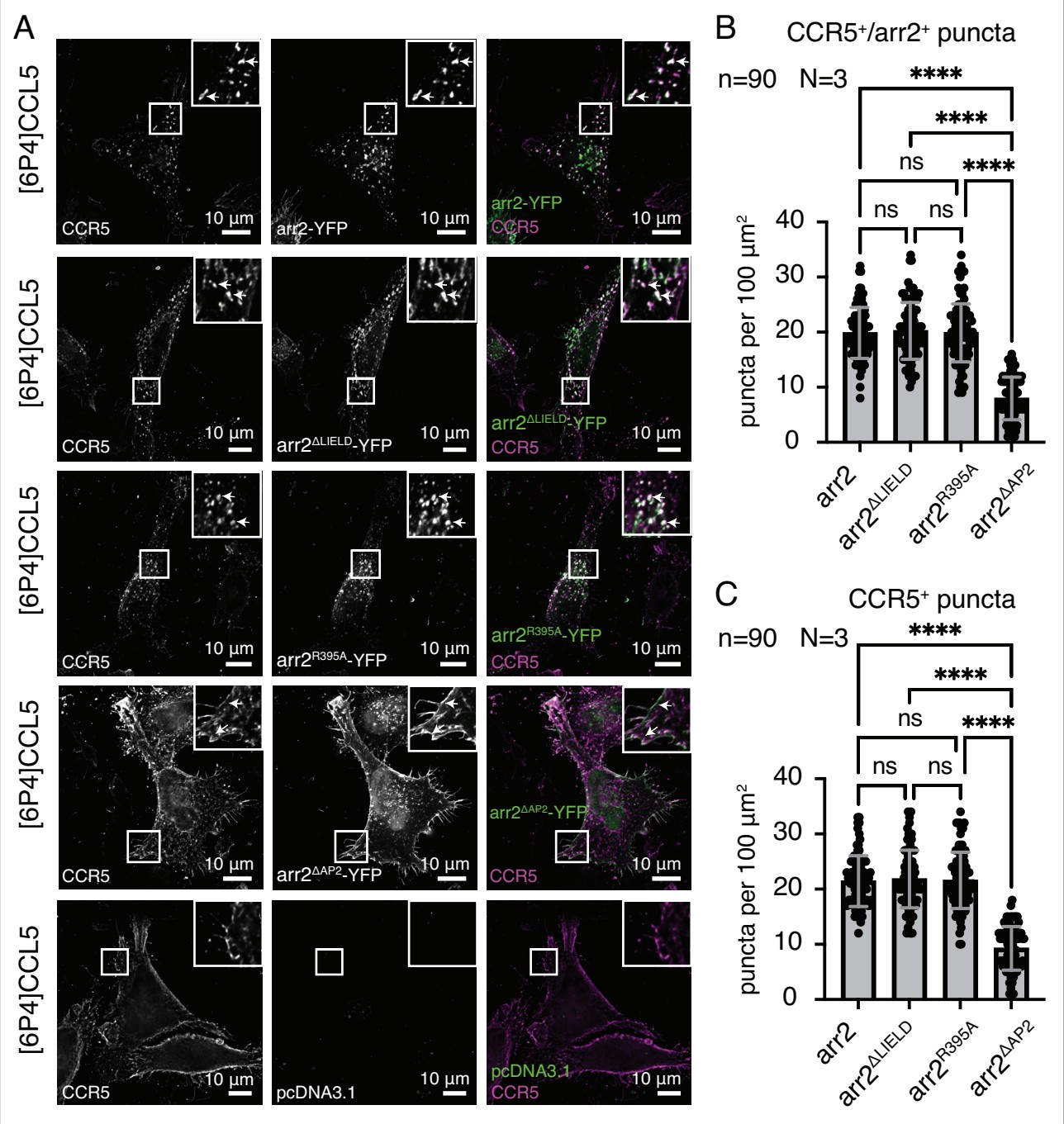

**Figure 5.** Dependence of CCR5 internalization on arrestin2 interactions with clathrin or AP2 monitored in HeLa[-arr2/3] cells. (**A**) CCR5 internalization induced by 60 min [6P4]CCL5 ligand stimulation in HeLa cells co-transfected with plasmids containing CCR5 and arrestin2-YFP, arrestin2-YFP[ΔLIELD], arrestin2-YFP[R395A], arrestin2-YFP[ΔAP2], or empty pcDNA3.1. (**B, C**) Internalization was quantified by counting the number of puncta (**B**) positive for CCR5 and arrestin2 (CCR5[+]/arrestin2[+]) or (**C**) only CCR5[+] 60 min after ligand stimulation in the HeLa[-arr2/3] cells. No significant difference in CCR5 internalization is detected for the arrestin2-YFP[ΔLIELD] and arrestin2-YFP[R395A] constructs (ns, not significant: $p>0.9999$), whereas the absence of the AP2 binding motif in the arrestin2[ΔAP2]-YFP construct causes a significant (****$p<0.0001$) reduction of CCR5 internalization 60 min after ligand incubation. Mean and standard deviation are shown for N=3 biological replicates and n=90 ROIs from 30 cells.

The online version of this article includes the following source data and figure supplement(s) for figure 5:

**Source data 1.** Values of CCR5+ and CCR5+/arr2+ puncta counting in HeLa[-arr2/3] cell line for *Figure 5B and C*.

**Figure supplement 1.** Dependence of CCR5 internalization on the interactions of arrestin2 with clathrin or AP2 monitored in HeLa (CCL2) cells by immunofluorescence microscopy.

*Figure 5 continued on next page*

*Figure 5 continued*

**Figure supplement 1—source data 1.** Values of Mander's colocalization coefficient, CCR5+/arr2+ and CCR5+ puncta counting in HeLa cells for arr2 vs. arr2^ΔLIELD mutant for *Figure 5—figure supplement 1A*.

**Figure supplement 1—source data 2.** Values of Mander's colocalization coefficient, CCR5+/arr2+ and CCR5+ puncta counting in HeLa cells for arr2 vs. arr2^ΔAP2 mutant for *Figure 5—figure supplement 1D*.

The mutation R395A within this motif has been found to inhibit β2AR internalization by arrestin3 (*Laporte et al., 2000*). Introducing the equivalent mutation into a fluorescent arrestin2 construct (arrestin2^R395A-YFP) did not lead to significant changes in CCR5 internalization relative to wild-type arrestin2-YFP (HeLa^-arr2/3: *Figure 5A–C*, HeLa: *Figure 5—figure supplement 1B*).

The structure of AP2β2 in complex with the C-terminal arrestin2 peptide (*Schmid et al., 2006*) shows arrestin2 residues D385, F388, F391, and R395 interacting via salt bridges and base stacking with AP2β2 (*Figure 3A*). Apparently, these interactions lead to a rather stable complex of full-length arrestin2 with AP2β2^701-937 in solution, as evidenced by our SEC binding assay, NMR, and the protection of activated arrestin2 against proteolysis (*Figure 3*, *Figure 3—figure supplement 3*). We hypothesized that due to this extended arrestin2-AP2β2 interaction interface, the mutation of a single interacting residue may not suffice to inhibit the arrestin2-AP2 interaction leading to CCR5 internalization. Hence, we generated a further arrestin2 mutant (arrestin2^ΔAP2-YFP) with all four direct interacting residues replaced (D385A, F388A, F391A, R395A). Similar to wild-type arrestin2-YFP, arrestin2^ΔAP2YFP is recruited to CCR5 at the plasma membrane (HeLa: *Figure 5—figure supplement 1C*). However, the subsequent CCR5 internalization is potently inhibited (HeLa^-arr2/3: *Figure 5A–C*, HeLa: *Figure 5—figure supplement 1D*) as indicated by a reduction of the Mander's CCR5/arrestin2 colocalization coefficient (from ~0.8 to ~0.2) the number of puncta positive for CCR5 and arrestin2 (from ~20–13 to ~8–6 per 100 μm², similar values obtained for CCR5 puncta). In agreement with this finding, the arrestin2^ΔAP2 mutant is unable to form a complex with AP2 in the presence of CCR5pp6 as assayed by SEC (*Figure 5—figure supplement 1E*). Taken together, these results show that the stable arrestin-AP2 interaction is crucial for CCR5 internalization, whereas the arrestin-clathrin interaction is not critical for sequestration from the plasma membrane.

## Discussion

In the present study, we have elucidated the role of CCR5 phosphorylation in arrestin-clathrin and arrestin-AP2 interactions which are essential for CCR5 sequestration into cells. NMR and chromatographic data on arrestin2 interactions with clathrin and AP2 proteins in the presence of arrestin2-activating CCR5 C-terminal phosphopeptides revealed the molecular details underlying CCR5 sequestration. Whereas the analysis of arrestin2 $^1$H-$^{15}$N resonances was difficult due to significant line broadening from intrinsic microsecond dynamics (*Isaikina et al., 2023*), the better quality of the clathrin-N $^1$H-$^{15}$N spectra allowed a detailed characterization of the arrestin2-clathrin interaction from the clathrin side. Arrestin2 binds to clathrin with about 100 μM affinity. This interaction is independent of arrestin2 activation by phosphopeptides, which may enable the internalization of receptors with low phosphorylation levels. The interaction is mediated through a single binding site in a 1:1 stoichiometry involving the arrestin2 clathrin-binding loop and the edges of blade 1 and blade 2 of clathrin-N. This binding site agrees with one of the binding sites observed in the crystal structure of the arrestin2-clathrin-N complex (PDB: 3GD1), but disagrees with the second crystal interaction site involving the arrestin2 splice loop. Presumably, the latter is a result of crystal packing. Our findings are supported by a previous study (*Lally et al., 2017*), which suggests that the arrestin C-domain edge containing the splice loop anchors arrestin to the membrane. The membrane anchoring seems mutually exclusive with clathrin binding (*Lally et al., 2017*).

In contrast to the arrestin2-clathrin interaction, the arrestin2-AP2 interaction requires arrestin2 activation by GPCR C-terminal tail phosphopeptides and depends quantitatively on their phosphorylation levels. The formed arrestin2-phosphopeptide-AP2 complexes are stable and more resistant to proteolysis than binary arrestin2-phosphopeptide complexes. The dissociation constant of the arrestin2-AP2 complex is about 10 μM in the presence of the fully phosphorylated CCR5pp6, which is a 10-fold higher affinity than that of the arrestin2-clathrin complex. This may indicate a preference towards AP2 complex formation for highly phosphorylated receptors. However, as the phosphorylation decreases,

**Figure 6.** Overall scheme of arr-class A and B GPCR internalization. Schematic difference of arrestin2-mediated internalization of arr-class B (left) vs. arr-class A (right) GPCRs. Arr-class B GPCRs bind stably to arrestin due to their high levels of phosphorylation. This results in a robust release of the arrestin C-terminus, a stable interaction with AP2, and formation of a long-lived GPCR•arrestin complex. Arr-class A GPCRs bind weakly to arrestin due to their poor phosphorylation. They require stabilization of the arrestin complex by membrane-bound PIP$_2$ molecules. The arrestin C-terminus is not fully released and consequently, the interaction with AP2 is unstable.

the arrestin2-AP2 dissociation constant weakens to about 40 µM, making it more comparable to the arrestin2-clathrin interaction.

Cellular experiments demonstrated chemokine agonist-induced recruitment of arrestin2 towards CCR5 at the plasma membrane, which is followed by the sequestration of the CCR5-arrestin2 complex into the cytoplasm, including the perinuclear region where it remains stable for an extended period. This provides experimental evidence that CCR5 belongs to the arr-class B of GPCRs as surmised previously from the large number of phosphorylation sites in its C-terminus and the presence of the pXpp motif (*Isaikina et al., 2023*). The sequestered receptor remained stable inside the cells for at least 30 min and was not rapidly degraded in lysosomes, even in the presence of [6P4]CCL5. A similar CCR5 lysosome-evading internalization pattern has been observed for the potent anti-HIV chemokine analog PSC-RANTES (*Escola et al., 2010*). As a recent CCR5 trafficking study demonstrated that CCR5 scavenges both CCL5 and [6P4]CCL5 chemokines in an arrestin-dependent manner (*Gu et al., 2023*), it is possible that the long retention of the CCR5-arrestin complex and the ability of the receptor to undergo recycling (*Gu et al., 2023*; *Mueller et al., 2002*) is utilized to remove the extracellular ligands from the cell surface. This may help in the regulation of chemokine ligand concentrations (*Ariel et al., 2006*).

Based on the present CCR5 data, we envision the following CME mechanism, which may also apply to other arr-class B receptors (*Figure 6*, left). The high receptor phosphorylation leads to a strong recruitment of arrestin even in the absence of interactions with negatively charged PIP$_2$ in the plasma membrane (*Janetzko et al., 2022*). The consequent tight binding of the phosphorylated receptor tail to arrestin strand β1 stably activates arrestin, thereby releasing its strand β20 and C-terminal tail. The freed β20 strand can then interact with the β2 subunit of AP2, which becomes released from the AP2 core upon AP2 activation by PIP$_2$ molecules (*Kelly et al., 2014*). Apparently, the interaction of arrestin2 strand β20 with the AP2 β2 subunit is rather strong, since it could only be inhibited in cellular assays by a fourfold, but not by a single mutation in the arrestin2 C-terminal region. In contrast,

removal of the clathrin-binding LIELD motif from arrestin2 did not impair receptor internalization, which is in agreement with the determined relatively low affinity of the arrestin2-clathrin interaction. This suggests that the clathrin interaction is less important for receptor internalization than the AP2 interaction.

The AP2 activation by $PIP_2$ also exposes the AP2 clathrin-binding LLNLD motif, responsible for clathrin recruitment and formation of the clathrin-coated pits (*Kelly et al., 2014*). In the stable ternary complex of a highly phosphorylated arr-class B receptor with arrestin and AP2, the exposed AP2 LLNLD clathrin-binding motif may then subsequently couple to the clathrin network, thereby inducing rapid sequestration from the plasma membrane into endosomes. The direct clathrin interaction of arrestin via its LIELD motif appears less important in this respect.

Although the generalization of this mechanism from CCR5 to other arr-class B receptors has to be explored further, it is indirectly corroborated in the visual rhodopsin-arrestin1 system. The arr-class B receptor rhodopsin (*Isaikina et al., 2023*) also undergoes CME (*Moaven et al., 2013*) with arrestin1 harboring the conserved AP2 binding motif, but missing the clathrin-binding motif (*Figure 1—figure supplement 1A*). As arr-class B receptors are almost all peptide-binding GPCRs (*Isaikina et al., 2023*; *Oakley et al., 2000*), we speculate that the strong arrestin interaction may also be utilized to remove the extracellular ligands from the plasma membrane, independent of receptor degradation.

In comparison to arr-class B receptors, arr-class A receptors have a lower phosphosite density in their C-terminal tails. Previous studies on the arr-class A β2AR (*Kang et al., 2009*; *Kim and Benovic, 2002*; *Laporte et al., 2000*) revealed that both clathrin and AP2 interaction are crucial for internalization. It is expected that the lower phosphosite density of arr-class A receptors results in a less stable interaction with arrestin (*Figure 6*, right). Therefore, additional interactions with $PIP_2$ lipids are required to anchor arrestin to the membrane and play a significant role in receptor endocytosis (*Janetzko et al., 2022*). As a consequence of the low phosphorylation of the receptor and the weakened interaction with arrestin strand β1, the equilibrium between inactive (strand β20 forms β-sheet with β1) and active (β20 released) arrestin is shifted towards the inactive conformation, even when arrestin is bound to the receptor via core contacts. Due to the low availability of strand β20, the arrestin interaction with AP2 is considerably weakened and comparable in strength to the clathrin interaction. This is consistent with the finding that AP2 and clathrin participate with similar strength in arr-class A receptor internalization (*Kang et al., 2009*; *Kim and Benovic, 2002*; *Laporte et al., 2000*). However, as both interactions are weak, the sequestration is reduced. In addition, the weaker receptor phosphorylation leads to fast dissociation of arrestin from the receptor at or near the plasma membrane, and in consequence, the receptor traffics without arrestin into endosomes (*Oakley et al., 2001*; *Oakley et al., 2000*).

In conclusion, this biophysical and cellular analysis of the interactions between the CCR5, arrestin2, and clathrin/AP2 endocytic machinery elucidates the interplay of molecular forces driving GPCR internalization. The strength of the receptor phosphorylation modulates the interaction with arrestin2, the release of its strand β20, and subsequently its interaction with the β2 subunit of AP2. In comparison, the arrestin2-clathrin interaction is weaker. The interaction between the receptor phosphosites and arrestin appears as a distinctive factor between arr-class B and arr-class A receptor sequestration, since phosphorylation of the latter is considerably weaker. These findings expand on the previously suggested mechanism for the reduced sequestration of the arr-class A receptor β2AR by *Kim and Benovic, 2002* and *Laporte et al., 2000*, where the weakened AP2-arrestin and the phosphorylation-independent clathrin-arrestin interactions appear to be of similar importance. The phosphorylation dependence of the AP2-arrestin interaction may also play a role in modulating the strength of downstream arrestin-mediated signaling from endosomes (*Jean-Charles et al., 2017*; *Tohgo et al., 2003*). A detailed understanding of these molecular interactions may provide a framework for the development of novel therapeutics interfering with GPCR sequestration.

## Materials and methods
### Constructs
The genes encoding full-length (arrestin2[1-418]) and truncated (arrestin2[1-393]) human arrestin2 constructs have been described before (*Isaikina et al., 2023*). The arrestin2[ΔCBL] construct in a pET-28a (+) vector was obtained from GenScript by replacing the loop residues of arrestin2 K357 to D385 with visual

arrestin residues E365 to N376. The construct also contained an N-terminal hexahistidine tag for purification, followed by a TEV cleavage site (ENLYFQG), resulting in the following total sequence (arrestin2$^{\Delta CBL}$ in bold, visual arrestin loop underlined):

MGSSHHHHHHSSGENLYFQG**MGDKGTRVFKKASPNGKLTVYLGKRDFVDHIDLVDPVDGVVLVD PEYLKERRVYVTLTCAFRYGREDLDVLGLTFRKDLFVANVQSFPPAPEDKKPLTRLQERLIKKLGEHA YPFTFEIPPNLPCSVTLQPGPEDTGKACGVDYEVKAFLAENLEEKIHKRNSVRLVIRKVQYAPERPGP QPTAETTRQFLMSDKPLHLEASLDKEIYYHGEPISVNVHVTNNTNKTVKKIKISVRQYADIVLFNTAQ YKVPVAMEEADDTVAPSSTFSKVYTLTPFLANNREKRGLALDGKLKHEDTNLASSTLLREGANREILG IIVSYKVKVKLVVSRGGLLGDLASSDVAVELPFTLMHPKP**<u>EDPAKESYQDAN</u>**IVFEDFARQRLKGMKD DKEEEEDGTGSPQLNNR**

clathrin-N [pETM33_CLTC_Clathrin propeller, clathrin heavy chain 1 residues 1–364, Addgene plasmid #178471, *Benz et al., 2022*] and AP2β2$^{701-937}$ [pETM33_AP2B1_adaptin, Addgene plasmid #178467, *Benz et al., 2022*] constructs were gifts from Ylva Ivarsson. Arrestin2-YFP was a gift from Robert Lefkowitz [Addgene plasmid # 36916, *Violin et al., 2006*].

The arrestin2$^{\Delta LIELD}$-YFP construct was obtained from the arrestin2-YFP template by standard PCR reaction using KAPA HiFi polymerase (forward primer: 5'-GAG ACT CCA GTA GAC ACC AAT ACC AAT GAT GAC GAC A-3'; reverse primer: 5'-TGT CGT CAT CAT TGG TAT TGG TGT CTA CTG GAG TCT C-3').

Arrestin2$^{R395A}$-YFP was generated in the same manner using forward primer: 5'-GGA CTT TGC TCG TCA GGC GCT GAA AGG CAT GAA G-3' and reverse primer: 5'-CTT CAT GCC TTT CAG CGC CTG ACG AGC AAA GTC C-3'.

The same protocol was applied for generating the arrestin2$^{\Delta AP2}$-YFP clone with the forward primer 5'-GAC ACC AAT GAT GAC GCC ATT GTG GCT GAG GAC GCT GCT CGT CAG GCG CTG AAA GGC ATG-3' and reverse primer 5'-CAT GCC TTT CAG CGC CTG ACG AGC AGC GTC CTC AGC CAC AAT GGC GTC ATC ATT GGT GTC-3'.

The CCR5 construct in pcDNA3.1 used in cellular assays was obtained from GenScript. The complete construct contained the full-length human CCR5 sequence (bold) followed by a GS-linker and a C-terminal FLAG-tag (underlined):

**MDYQVSSPIYDINYYTSEPCQKINVKQIAARLLPPLYSLVFIFGFVGNMLVILILINCKRLKSMTDIY LLNLAISDLFFLLTVPFWAHYAAAQWDFGNTMCQLLTGLYFIGFFSGIFFIILLTIDRYLAVVHAVFA LKARTVTFGVVTSVITWVVAVFASLPGIIFTRSQKEGLHYTCSSHFPYSQYQFWKNFQTLKIVILGLV LPLLVMVICYSGILKTLLRCRNEKKRHRAVRLIFTIMIVYFLFWAPYNIVLLLNTFQEFFGLNNCSSS NRLDQAMQVTETLGMTHCCINPIIYAFVGEKFRNYLLVFFQKHIAKRFCKCCSIFQQEAPERASSVYT RSTGEQEISVGL**GSGGGGSGGGSSSGGGGSGGGSSSGGEF<u>DYKDDDDK</u>

The genes encoding [5P12]CCL5 (*Wiktor et al., 2013*), CCL5 (*Wiktor et al., 2013*) and [6P4]CCL5 (*Isaikina et al., 2022*) for expression in *E. coli* have been described previously.

## Protein expression and purification

All protein constructs were expressed in *E. coli* BL21 (DE3) cultured in Lysogeny broth medium (LB). For the preparation of the deuterated $^{15}$N-labeled (and $^{13}$C-labeled) NMR samples, cells were grown in D$_2$O/$^{15}$NH$_4$Cl ($^{13}$C-glucose) M9 minimal medium. For the preparation of Ile-δ1-$^{13}$CH$_3$-labeled arrestin2$^{1-393}$, the minimal medium was supplemented with 80 mg per liter of 2-ketobutyric acid-4-$^{13}$C,3,3-d$_2$ sodium salt hydrate following previously reported procedures (*Tugarinov et al., 2006*). In all cases, cells were grown at 37°C until the optical density at 600 nm (OD$_{600\ nm}$) reached 0.7–0.8. Thereafter, protein expression was induced by the addition of 25 μM isopropyl β-D-thiogalactopyranoside (IPTG) (arrestin2 constructs) or 1 mM IPTG (clathrin-N and AP2β2$^{701-937}$). The temperature was lowered to 18°C (arrestin2 constructs) or 22°C (clathrin-N and AP2β2$^{701-937}$) for an overnight expression, after which the cells were harvested by centrifugation (30 min at 6000 rpm).

### Arrestin2

The purification of arrestin2 has been described before (*Isaikina et al., 2023*). In short, arrestin2$^{1-418}$, arrestin2$^{1-393}$, arrestin2$^{\Delta CBL}$ and arrestin2$^{\Delta AP2}$ were purified on a Ni-NTA HiTrap HP column (GE Life Sciences) and the His-tag was removed by overnight cleavage with TEV protease (homemade). The cleaved protein was further separated from impurities by a reverse IMAC step on a Ni-NTA HiTrap HP column. An ion exchange chromatography step using a 5 ml HiTrap Q HP anion exchange column was

then carried out before the final gel filtration step on a HiLoad 16/600 Superdex 200 pg gel filtration column (GE Healthcare) equilibrated with 20 mM HEPES, 150 mM NaCl, pH 7.4 (SEC buffer I). Protein purity was confirmed by SDS-PAGE.

### clathrin-N

clathrin-N was purified similarly to the arrestin2 protocol using a Ni-NTA HiTrap HP column. The His-tag was removed by overnight cleavage with 3C protease (homemade), followed by a reverse IMAC step. The protein was concentrated in a Vivaspin 20 (Sartorius) concentrator (10 kDa MWCO) before a final gel filtration step on a HiLoad 16/600 Superdex S75 pg gel filtration column (GE Healthcare) equilibrated with SEC buffer I. Protein purity was confirmed by SDS-PAGE.

### AP2β2

AP2β2 was purified similarly to clathrin-N. Due to a change of the isoelectric point from 6.6 to 7.7 after 3C protease cleavage, the pH of the SEC buffer was changed to 7.0 (SEC buffer II, 20 mM HEPES, 150 mM NaCl, pH 7.0).

### Determination of protein melting temperature

Protein melting temperatures were determined routinely by differential scanning fluorimetry using a Prometheus Nanotemper NT.48 (Tecan) instrument.

## Peptide synthesis

Phosphorylated peptides (*Supplementary file 1*) corresponding to the last 22 residues of the human CCR5 receptor (CCR5pp3, CCR5pp4, CCR5pp6) were obtained from the Tufts University Core Facility for peptide synthesis.

## Cell lines

The HeLa (ATCC CCL2) cell line was a kind gift from Prof. Martin Spiess. These standard cell lines were authenticated by ATCC. In addition, more recently (2021), the cell lines' identities were authenticated by STR analysis by Microsynth AG (Switzerland) and tested negative for mycoplasma contamination.

The HeLa KO cell line (HeLa$^{-arr2/3}$) was generated from HeLa cells and was a kind gift from Prof. Mariagrazia Uguccioni and Dr. Valentina Cecchinato (Institute for Research in Biomedicine, Bellinzona, Switzerland). Amplification of DNA genomic products and Sanger genomic sequencing were performed by Microsynth AG (Balgach, Switzerland).

## Cell culture

HeLa (ATCC CCL2) cells and HeLa KO cells (HeLa$^{-arr2/3}$) (*D'Agostino et al., 2020*) devoid of arrestin2 and arrestin3 were grown in high-glucose Dulbecco's modified Eagle's medium (Sigma-Aldrich) with 10% fetal bovine serum (FBS, Biowest), 2 mM L-glutamine (#25030; Gibco), 100 U ml$^{-1}$ penicillin G and 100 ng ml$^{-1}$ streptomycin (P433; Sigma-Aldrich Chemie GmbH), 1 mM sodium pyruvate (#S8636; Sigma-Aldrich Chemie GmbH) at 37°C and 5% CO$_2$. For transient cell transfections, cells were plated into six-well plates to reach 70% confluency the following day and transfected with 0.3 μg of each arrestin2-YFP and CCR5 plasmid DNA complexed with Helix-IN transfection reagent (OZ Biosciences).

## Immunostaining and microscopy

In HeLa cell lines, arrestin2 and CCR5 were overexpressed by transiently transfecting arrestin2-YFP and CCR5, respectively. CCR5 was stained with a primary FLAG antibody (1:200, Thermo Fisher F3165) in 5% FBS in PBS. A secondary mouse (1:500) Alexa 633 (A21052, Invitrogen) antibody was used in 5% FBS in PBS. Fluoromount-G mounting medium (Thermo Fischer) was used to mount the coverslips onto the imaging slide. Fluorescence and DIC images were acquired with an ORCA-Flash 4.0 camera (Hamamatsu) mounted on an Axio Imager.M2 fluorescence microscope with a 63x Plan-Apochromat objective (Carl Zeiss) and an HXP 120C light source with ZEN 2.6 software. Image processing was performed using the OMERO.insight client (*Allan et al., 2012*) and analyzed with Fiji software (*Schindelin et al., 2012*).

To assess the recruitment of arrestin2 by CCR5, arrestin2-YFP- and CCR5-expressing cells were incubated with 1 µM chemokine ligands. Individual coverslips were removed and fixed with 4% PFA for 10 min at 0, 10, and 30 min post-ligand stimulation. Fixed cells were permeabilized with 0.1% Triton X-100 in PBS for 5 min, rinsed with PBS twice, and blocked with 5% FBS in PBS. CCR5 was stained as mentioned previously and imaged. Colocalization between arrestin2 and CCR5 was assessed by measuring the Mander's coefficient in 2–3 regions of interest (ROI) of 100 µm$^2$ per cell using the Fiji JACOP plug-in (*Bolte and Cordelières, 2006*). The number of internalized CCR5 puncta with and without arrestin2 was quantified manually by taking 3 ROIs of 100 µm$^2$ per cell using Fiji.

To image arrestin2/CCR5 trafficking to lysosomes, arrestin2-YFP and CCR5-expressing cells were treated with [6P4]CCL5. Individual coverslips were removed and fixed with 4% PFA for 10 min at 0, 10, and 30 min post-stimulation. Fixed cells were permeabilized with 0.1% Triton X-100 in PBS for 5 min, rinsed with PBS twice, and blocked with 5% FBS in PBS. Lysosomes were stained with LAMP1 antibody (1:200, Cell Signaling Technology, D2D11). CCR5 was stained with FLAG antibody (1:200, Thermo Fisher F3165). Secondary rabbit (1:500) Alexa-Fluor 568 (A110042 Invitrogen) and secondary mouse (1:500) Alexa-Fluor 633 (A21052, Invitrogen) fluorescent antibodies were used, respectively. Fixation, imaging, and analysis were performed as described above.

For assessing the effect of arrestin2 mutations, respective mutants were overexpressed with CCR5 in HeLa$^{-arr2/3}$ and HeLa cells, followed by immunostaining and microscopy analysis as described above (HeLa$^{-arr2/3}$: 60 min, HeLa: 30 min post [6P4]CCL5 stimulation).

## Live-cell imaging

For live-cell imaging, HeLa cells expressing arrestin2-YFP were seeded on an ibidi µ-slide VI 0.4 channel slide (#80606, Ibidi), 24 hr prior to data acquisition. Live cell imaging was performed in complete growth medium lacking phenol red at 37°C with 5% CO$_2$ 508 using an inverted Axio Observer microscope (Zeiss) with a Plan Apochromat N 63x/1.40 oil DIC 509 M27 objective and a Photometrics Prime 95B camera. Filters with standard specifications for GFP were used to image arrestin2-YFP. After selecting cells to be imaged, images were acquired at 5 min intervals. Image processing was performed using the OMERO.insight client (*Allan et al., 2012*) and analyzed with Fiji software (*Schindelin et al., 2012*).

To image lysosome stained with lysotracker, HeLa cells expressing arrestin2-YFP and CCR5 were seeded on an ibidi µ-slide VI 0.4 channel slide (#80606, Ibidi), 24 hr prior to data acquisition. LysoTracker Deep Red (L12492) and peptide ligand was added to complete growth medium lacking phenol red. Images were taken at 0 min and 30 min.

## Statistics

Statistical analysis was performed using GraphPad Prism 10.1.1 (GraphPad Software, Inc, San Diego, CA, USA). For statistical evaluation, the normality of the data was routinely tested using the Shapiro-Wilk or Kolmogorov-Smirnov normality test. For the comparison of two groups (*Figure 5A and B*), a t-test was employed, whereas to compare three or more groups (*Figure 4D*), a one-way ANOVA was performed followed by a Kruskal-Wallis test.

## NMR experiments

NMR samples were prepared in SEC buffer I supplemented with 5% D$_2$O, 2 mM EDTA, and 0.03% NaN$_3$ as 270 µl volumes in Shigemi tubes. The assignment sample contained 300 mM NaCl. The spectra were recorded on a Bruker AVANCE 14.1-Tesla (600 MHz) spectrometer (arrestin2$^{1-393}$, AP2β2$^{701-937}$) or a Bruker AVANCE 21.2-Tesla (900 MHz) spectrometer (clathrin-N) equipped with a TCI cryoprobe at 303 K. The spectra were processed with NMRPipe (*Delaglio et al., 1995*), assigned using CcpNmr AnalysisAssign (*Skinner et al., 2016*), and analyzed with NMRFAM-SPARKY (*Lee et al., 2015*).

### Arrestin2 assignment

For the assignment of arrestin2 backbone resonances, HNCACB-TROSY, HNCA-TROSY, and HNCO-TROSY spectra were recorded on $^{15}$N-, $^{13}$C-, and $^2$H-labeled arrestin2$^{1-393}$ (100-150 µM) samples.

## Arrestin2 titrations

For detecting the clathrin-N-arrestin2 interaction on arrestin2 by NMR, a titration of 50 µM $^{15}$N-, $^{2}$H-labeled arrestin2$^{1-393}$ with unlabeled clathrin-N up to an equimolar amount was monitored by $^{1}$H-$^{15}$N HSQC-TROSY spectra recorded as 100 ($^{15}$N) × 512 ($^{1}$H) complex points and acquisition times of 50 ms ($^{15}$N) and 55 ms ($^{1}$H) for 4.5 hr. Peak intensities $I$ were determined by integration, and the interaction was quantified relative to the intensity $I_0$ in the absence of clathrin-N. An intensity attenuation larger than one standard deviation of $I/I_0$ calculated from all resonances was considered significant.

Similarly, a titration of 50 µM Ile-δ1-$^{13}$CH$_3$, $^{2}$H-labeled arrestin2$^{1-393}$ with unlabeled clathrin-N up to a concentration of 600 µM was monitored by $^{1}$H-$^{13}$C HMQC spectra recorded as 32 ($^{13}$C) × 512 ($^{1}$H) complex points and acquisition times of 20 ms ($^{13}$C) and 55 ms ($^{1}$H) for 1 hr. The assignments of the methyl groups were transferred from BMRB ID:51131 (*Shiraishi et al., 2021*). K$_D$ values were obtained by nonlinear least-squares fitting using Matlab (Matlab_R2021b, MathWorks, Inc) and the following equation:

$$\Delta\delta = \Delta\delta_{max} \frac{[L] + [P] + K_D - \sqrt{\left([L] + [P] + K_D\right)^2 - 4\,[P]\,[L]}}{2\,[P]}$$

where $\Delta\delta = \delta_{apo}\text{-}\delta_{bound}$ is the difference in the $^{13}$C chemical shift, $\delta_{max}$ is the difference between apo and the fully ligand-bound state, and [P] and [L] are the total protein (arrestin2) and ligand (clathrin-N) concentrations, respectively.

Combined $^{1}$H/$^{13}$C chemical shift perturbations $\Delta\delta_{HC}$ were calculated from the relation:

$$\Delta\delta_{HC} = \sqrt{\Delta\delta_H{}^2 + \left(\frac{\Delta\delta_C}{3.6}\right)^2} \tag{2}$$

where $\Delta\delta_H$ is the methyl proton and $\Delta\delta_C$ is the $^{13}$C-methyl chemical shift difference between the apo and bound state. $\Delta\delta_{HC}$ values greater than one standard deviation from the mean were considered significant.

## clathrin-N titrations

For detecting the clathrin-N-arrestin2 interaction on clathrin-N by NMR, a titration of 50 µM $^{15}$N-, $^{2}$H-labeled clathrin-N with unlabeled arrestin2$^{1-393}$, arrestin2$^{1-418}$, arrestin2$^{\Delta CBL}$ or arrestin2$^{1-393}$•CCR5pp6 up to an equimolar amount was monitored by $^{1}$H-$^{15}$N HSQC-TROSY spectra recorded as 100 ($^{15}$N) × 1024 ($^{1}$H) complex points and acquisition times of 35 ms ($^{15}$N) and 80 ms ($^{1}$H) for 4.5 hr. Assignments for clathrin-N were transferred from BMRB ID:25403 (*Zhuo et al., 2015*). Peak intensities were determined by quantifying peak volumes. Binding regions were determined by intensity ratios as described above.

## AP2β2 titrations

For detecting the AP2β2-arrestin2 interaction on AP2β2$^{701-937}$ by NMR, a titration of 50 µM $^{15}$N-, $^{2}$H-labeled AP2β2$^{701-937}$ with unlabeled arrestin2$^{1-418}$ or arrestin2$^{1-418}$•CCR5pp6 up to an equimolar amount was monitored by $^{1}$H-$^{15}$N HSQC-TROSY spectra recorded as 100 ($^{15}$N) × 512 ($^{1}$H) complex points and acquisition times of 50 ms ($^{15}$N) and 55 ms ($^{1}$H) for 1 hr. Assignments for AP2β2$^{701-937}$ were transferred from BMRB ID: 52320 (*Naudi-Fabra et al., 2024*). Peak intensities were determined by quantifying peak volumes. Binding regions were determined by intensity ratios as described above.

## SEC binding assay

SEC binding assays were performed on an UltiMate 3000 HPLC system. Samples containing single polypeptides or their mixtures, as indicated in the text and figure legends (arrestin2$^{1-418}$: 10 µM, AP2β2: 10 µM, clathrin-N: 10 µM, CCR5pp0: 100 µM, CCR5pp3: 100 µM, CCR5pp4: 100 µM, CCR5pp6: 25–100 µM) were prepared as 270 µL volumes in SEC buffer II. Samples were injected with an autosampler using a 250 µL sample loop onto a self-packed 4.2-ml S200 10/300 SEC column (length 25 mm, diameter 4.6 mm), which had been pre-equilibrated with SEC buffer II. Complex formation was monitored by following protein absorbance at 280 nm at 2.6 ml elution volume and the composition of

the respective SEC fractions of all prepared complexes was analyzed by SDS-PAGE (*Figure 3—figure supplement 2C*).

The apparent dissociation constant of the arrestin2$^{1-418}$-AP2β2$^{701-937}$ complex in the presence of the CCR5 phosphopeptides could be determined by integrating the SEC absorption peaks of the complex and of the apo proteins and using their respective extinction coefficients. The increase in absorbance around 2.6 ml is a consequence of the additional AP2β2$^{701-937}$ in that elution region. This region overlaps to some extent with the elution profile of apo arrestin2$^{1-418}$. Taking into account the 1:1 stoichiometry, the amount of formed complex was estimated by integrating the profile between 2.4 and 2.8 ml, subtracting the respective integral of the apo arrestin2$^{1-418}$ profile, and dividing the difference by the AP2β2$^{701-937}$ extinction coefficient. Since the phosphopeptides do not contain tryptophan, their absorbance contribution was ignored.

## Trypsin proteolysis

Trypsin proteolysis assays were carried out by pre-incubating 100 µL of 20 µM arrestin2$^{1418}$ and 100 µM CCR5pp6 or 500 µM CCR5pp4/CCR5pp3 phosphopeptide with 1- (CCR5pp6), 2- (CCR5pp6) and 5-molar equivalents (CCR5pp6, CCR5pp4, CCR5pp3) of AP2β2 protein at room temperature for 5–10 min in SEC buffer II. Following pre-incubation, 1 ng of Trypsin Gold (Promega) was added, and the mixture was further incubated at 35°C under gentle shaking (500 rpm, ThermoMixer). Samples for SDS-PAGE (10 µL mixed with 10 µL of 4x SDS loading buffer) were taken at 0, 5, 10, 20, 30, 60, and 90 min. The reaction was quenched by boiling the samples for 10 min at 95°C. Samples were then loaded on 4–20% precast gradient gels and visualized with Instant Blue protein stain (Abcam). Control reactions were run with apo arrestin2$^{1-418}$ and arrestin2$^{1-418}$ incubated with 100 µM CCR5pp6 or 60–500 µM CCR5pp4/CCR5pp3 phosphopeptide.

SDS-PAGE gels were analyzed using ImageJ software (*Schneider et al., 2012*) by integrating the peak area of the two bands corresponding to full-length arrestin2$^{1-418}$ (top) and arrestin2$^{1-418}$ cleaved at residue R393 (bottom). The relative abundance (in %) of arrestin2$^{1-418}$ was calculated from the respective integrals at 5 min upon trypsin addition.

## Acknowledgements

We gratefully acknowledge I Hertel and M Rogowski for the expression and purification of proteases and CCL5 ligands. We thank Prof. Mariagrazia Uguccioni and Dr. Valentina Cecchinato (Institute for Research in Biomedicine, Bellinzona, Switzerland) for the HeLa$^{-arr2/3}$ cells and Prof. Arun Shukla for helpful discussions. We further acknowledge the Biophysics Facility at the Biozentrum, University of Basel, for their support and assistance with the biophysical characterization of arrestin2. This work was supported by the Swiss National Science Foundation (grants 31–201270 and IZLIZ3-200298 to SG, 31–197779 and 31–185127 to AS), and by a Fellowship for Excellence by the Biozentrum Basel International PhD Program to IP.

## Additional information

### Funding

| Funder | Grant reference number | Author |
| --- | --- | --- |
| Schweizerischer Nationalfonds zur Förderung der Wissenschaftlichen Forschung | 31-201270 | Stephan Grzesiek |
| Schweizerischer Nationalfonds zur Förderung der Wissenschaftlichen Forschung | IZLIZ3-200298 | Stephan Grzesiek |

| Funder | Grant reference number | Author |
|---|---|---|
| Schweizerischer Nationalfonds zur Förderung der Wissenschaftlichen Forschung | 31-197779 | Anne Spang |
| Schweizerischer Nationalfonds zur Förderung der Wissenschaftlichen Forschung | 31-185127 | Anne Spang |
| Fellowship for Excellence by the Biozentrum Basel International PhD Program | | Ivana Petrovic |

The funders had no role in study design, data collection and interpretation, or the decision to submit the work for publication.

## Author contributions

Ivana Petrovic, Conceptualization, Data curation, Formal analysis, Validation, Investigation, Visualization, Methodology, Writing – original draft, Project administration, Writing – review and editing, IP conceived the study. IP generated constructs and mutants for the cellular assays. IP expressed and purified proteins. IP recorded NMR experiments and analyzed NMR data. IP designed and analyzed SEC and trypsin proteolysis assay. IP wrote the manuscript with inputs from all co-authors; Samit Desai, Conceptualization, Data curation, Formal analysis, Validation, Investigation, Visualization, Methodology, Writing – original draft, Writing – review and editing, SD conceived the study. SD performed the cellular assays and analyzed the cellular data. SD participated in manuscript writing; Polina Isaikina, Conceptualization, Writing – original draft, PI conceived the study. PI provided writing inputs; Layara Akemi Abiko, Conceptualization, Formal analysis, Investigation, Writing – original draft, Writing – review and editing, LAA generated constructs and mutants for the cellular assays. LAA provided manuscript writing inputs; Anne Spang, Conceptualization, Data curation, Supervision, Funding acquisition, Methodology, Writing – original draft, Project administration, Writing – review and editing, AS conceived the study. AS provided guidance for the cellular assays. AS provided inputs and participated in manuscript writing; Stephan Grzesiek, Conceptualization, Data curation, Formal analysis, Supervision, Funding acquisition, Investigation, Visualization, Methodology, Writing – original draft, Project administration, Writing – review and editing, SG conceived the study. SG wrote the manuscript with inputs from all co-authors

## Author ORCIDs

Ivana Petrovic ⓘ https://orcid.org/0000-0003-4461-7603
Samit Desai ⓘ https://orcid.org/0009-0002-4175-6659
Polina Isaikina ⓘ https://orcid.org/0000-0003-4466-7715
Layara Akemi Abiko ⓘ https://orcid.org/0000-0003-3537-534X
Anne Spang ⓘ https://orcid.org/0000-0002-2387-6203
Stephan Grzesiek ⓘ https://orcid.org/0000-0003-1998-4225

Reviewer #1 (Public review): https://doi.org/10.7554/eLife.106839.4.sa1
Reviewer #2 (Public review): https://doi.org/10.7554/eLife.106839.4.sa2
Reviewer #3 (Public review): https://doi.org/10.7554/eLife.106839.4.sa3
Author response https://doi.org/10.7554/eLife.106839.4.sa4

# Additional files

## Supplementary files

Supplementary file 1. CCR5 phosphopeptide sequences and their affinities towards arrestin2. Underlined serine or threonine residues are phosphorylated.

MDAR checklist

## Data availability

All data needed to evaluate the conclusions in the paper are present in the main text and/or the Supplementary Information. All materials used in this study are available from the corresponding authors upon reasonable request.

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
