## [Editor Report · eLife Assessment]

The authors investigate arrestin2-mediated CCR5 endocytosis in the context of clathrin and AP2 contributions. Using an extensive set of NMR experiments, and supported by microscopy and other biophysical assays, the authors provide **compelling** data on the roles of AP2 and clathrin in CCR5 endocytosis. This **important** work will appeal to an audience beyond those studying chemokine receptors, including those studying GPCR regulation and trafficking. The distinct role of AP2 and not clathrin will be of particular interest to those studying GPCR internalization mechanisms.

---

## [Referee Report · Reviewer #1 (Public review)]

Petrovic et al. investigate CCR5 endocytosis via arrestin2, with a particular focus on clathrin and AP2 contributions. The study is thorough and methodologically diverse. The NMR titration data clearly demonstrate chemical shift changes at the canonical clathrin-binding site (LIELD), present in both the 2S and 2L arrestin splice variants. To assess the effect of arrestin activation on clathrin binding, the authors compare: truncated arrestin (1-393), full-length arrestin, and 1-393 incubated with CCR5 phosphopeptides. All three bind clathrin comparably, whereas controls show no binding. These findings are consistent with prior crystal structures showing peptide-like binding of the LIELD motif, with disordered flanking regions. The manuscript also evaluates a non-canonical clathrin binding site specific to the 2L splice variant. Though this region has been shown to enhance beta2-adrenergic receptor binding, it appears not to affect CCR5 internalization.

Similar analyses applied to AP2 show a different result. AP2 binding is activation-dependent and influenced by the presence and level of phosphorylation of CCR5-derived phosphopeptides. These findings are reinforced by cellular internalization assays.

In sum, the results highlight splice-variant-dependent effects and phosphorylation-sensitive arrestin-partner interactions. The data argue against a (rapidly disappearing) one-size-fits-all model for GPCR-arrestin signaling and instead support a nuanced, receptor-specific view, with one example summarized effectively in the mechanistic figure.

---

## [Referee Report · Reviewer #2 (Public review)]

Summary:

Based on extensive live cell assays, SEC, and NMR studies of reconstituted complexes, these authors explore the roles of clathrin and the AP2 protein in facilitating clathrin mediated endocytosis via activated arrestin-2. NMR, SEC, proteolysis, and live cell tracking confirm a strong interaction between AP2 and activated arrestin using a phosphorylated C-terminus of CCR5. At the same time a weak interaction between clathrin and arrestin-2 is observed, irrespective of activation.

These results contrast with previous observations of class A GPCRs and the more direct participation by clathrin. The results are discussed in terms of the importance of short and long phosphorylated bar codes in class A and class B endocytosis.

Strengths:

The 15N,1H and 13C,methyl TROSY NMR and assignments represent a monumental amount of work on arrestin-2, clathrin, and AP2. Weak NMR interactions between arrestin-2 and clathrin are observed irrespective of activation of arrestin. A second interface, proposed by crystallography, was suggested to be a possible crystal artifact. NMR establishes realistic information on the clathrin and AP2 affinities to activated arrestin with both kD and description of the interfaces.

---

## [Referee Report · Reviewer #3 (Public review)]

Summary:

Overall, this is a well-done study, and the conclusions are largely supported by the data, which will be of interest to the field.

Strengths:

Strengths of this study include experiments with solution NMR that can resolve high-resolution interactions of the highly flexible C-terminal tail of arr2 with clathrin and AP2. Although mainly confirmatory in defining the arr2 CBL 376LIELD380 as the clathrin binding site, the use of the NMR is of high interest (Fig. 1). The 15N-labeled CLTC-NTD experiment with arr2 titrations reveals a span from 39-108 that mediates an arr2 interaction, which corroborates previous crystal data, but does not reveal a second area in CLTC-NTD that in previous crystal structures was observed to interact with arr2.

SEC and NMR data suggest that full-length arr2 (1-418) binding with 2-adaptin subunit of AP2 is enhanced in the presence of CCR5 phospho-peptides (Fig. 3). The pp6 peptide shows the highest degree of arr2 activation, and 2-adaptin binding, compared to less phosphorylated peptide or not phosphorylated at all. It is interesting that the arr2 interaction with CLTC NTD and pp6 cannot be detected using the SEC approach, further suggesting that clathrin binding is not dependent on arrestin activation. Overall, the data suggest that receptor activation promotes arrestin binding to AP2, not clathrin, suggesting the AP2 interaction is necessary for CCR5 endocytosis.

To validate the solid biophysical data, the authors pursue validation experiments in a HeLa cell model by confocal microscopy. This requires transient transfection of tagged receptor (CCR5-Flag) and arr2 (arr2-YFP). CCR5 displays a "class B"-like behavior in that arr2 is rapidly recruited to the receptor at the plasma membrane upon agonist activation, which forms a stable complex that internalizes onto endosomes (Fig. 4). The data suggest that complex internalization is dependent on AP2 binding not clathrin (Fig. 5).

The addition of the antagonist experiment/data adds rigor to the study.

Overall, this is a solid study that will be of interest to the field.

---

## [Author Response]

The following is the authors’ response to the previous reviews

**Public Reviews:**

**Reviewer #1 (Public review):**
Petrovic et al. investigate CCR5 endocytosis via arrestin 2, with a particular focus on clathrin and AP2 contributions. The study is thorough and methodologically diverse. The NMR titration data clearly demonstrate chemical shift changes at the canonical clathrin-binding site (LIELD), present in both the 2S and 2L arrestin splice variants.To assess the effect of arrestin activation on clathrin binding, the authors compare: truncated arrestin (1-393), full-length arrestin, and 1-393 incubated with CCR5 phosphopeptides. All three bind clathrin comparably, whereas controls show no binding. These findings are consistent with prior crystal structures showing peptide-like binding of the LIELD motif, with disordered flanking regions. The manuscript also evaluates a non-canonical clathrin binding site specific to the 2L splice variant. Though this region has been shown to enhance beta2-adrenergic receptor binding, it appears not to affect CCR5 internalization.Similar analyses applied to AP2 show a different result. AP2 binding is activation-dependent and influenced by the presence and level of phosphorylation of CCR5-derived phosphopeptides. These findings are reinforced by cellular internalization assays.In sum, the results highlight splice-variant-dependent effects and phosphorylation-sensitive arrestin-partner interactions. The data argue against a (rapidly disappearing) one-size-fitsall model for GPCR-arrestin signaling and instead support a nuanced, receptor-specific view, with one example summarized effectively in the mechanistic figure.

We thank the referee for this positive assessment of our manuscript. Indeed, by stepping away from the common receptor models for understanding internalization (b2AR and V2R), we revealed the phosphorylation level of the receptor as a key factor in driving the sequestration of the receptor from the plasma membrane. We hope that the proposed mechanistic model will aid further studies to obtain an even more detailed understanding of forces driving receptor internalization.

Weaknesses:Figure 1 shows regions alphaFold model that are intrinsically disordered without making it clear that this is not an expected stable position. The authors NMR titration data are n=1. Many figure panels require that readers pinch and zoom to see the data.

In the “Recommendations for the Authors” section, we addressed the reviewer’s stated weaknesses. In short, for the AlphaFold representation in Figure 1A, we added explicit labeling and revised the legend and main text to clearly state that the depicted loops are intrinsically disordered, absent from crystal structures due to flexibility, and shown only for visualization of their location. We also clarified that the NMR titration experiments inherently have n = 1 due to technical limitations, and that this is standard practice in the field, while ensuring individual data points remain visible. The supplementary NMR figures now have more vibrant coloring, allowing easier data assessment. However, we have not changed the zooming of the microscopy and NMR spectra. We believe that the presentation of microscopy data, which already show zoomed-in regions of interest, follow standard practices in the field. Furthermore, we strongly believe that we should display full NMR spectra in the supplementary figures to allow the reader to assess the overall quality and behavior. As indicated previously, the reader can zoom in to very high resolution, since the spectra are provided by vector graphics. Zoomed regions of the relevant details are provided in the main figures.

**Reviewer #2 (Public review):**
Summary:Based on extensive live cell assays, SEC, and NMR studies of reconstituted complexes, these authors explore the roles of clathrin and the AP2 protein in facilitating clathrin mediated endocytosis via activated arrestin-2. NMR, SEC, proteolysis, and live cell tracking confirm a strong interaction between AP2 and activated arrestin using a phosphorylated C-terminus of CCR5. At the same time a weak interaction between clathrin and arrestin-2 is observed, irrespective of activation.These results contrast with previous observations of class A GPCRs and the more direct participation by clathrin. The results are discussed in terms of the importance of short and long phosphorylated bar codes in class A and class B endocytosis.Strengths:The 15N,1H and 13C,methyl TROSY NMR and assignments represent a monumental amount of work on arrestin-2, clathrin, and AP2. Weak NMR interactions between arrestin-2 and clathrin are observed irrespective of activation of arrestin. A second interface, proposed by crystallography, was suggested to be a possible crystal artifact. NMR establishes realistic information on the clathrin and AP2 affinities to activated arrestin with both kD and description of the interfaces.

We sincerely thank the referee for this encouraging evaluation of our work and appreciate the recognition of the NMR efforts and insights into the arrestin–clathrin–AP2 interactions.

Weaknesses:This reviewer has identified only minor weaknesses with the study.(1) I don't observe two overlapping spectra of Arrestin2 (1393) +/- CLTC NTD in Supp Figure 1

We believe the referee is referring to Figure 1 – figure supplement 2. We have now made the colors of the spectra more vibrant and used different contouring to make the differences between the two spectra clearer. The spectra are provided as vector graphics, which allows zooming in to the very fine details.

(2) Arrestin-2 1-418 resonances all but disappear with CCR5pp6 addition. Are they recovered with Ap2Beta2 addition and is this what is shown in Supp Fig 2D

We believe the reviewer is referring to Figure 3 - figure supplement 1. In this figure, the panels E and F show resonances of arrestin2^1-418^ (apo state shown with black outline) disappear upon the addition of CCR5pp6 (arrestin2^1-418^•CCR5pp6 complex spectrum in red). The panels C and D show resonances of arrestin2^1-418^ (apo state shown with black outline), which remain unchanged upon addition of AP2b2 ^701-937^ (orange), indicating no complex formation. We also recorded a spectrum of the arrestin2^1-418^•CCR5pp6 complex under addition of AP2b2 ^701-937^ (not shown), but the arrestin2 resonances in the arrestin2^1-418^ •CCR5pp6 complex were already too broad for further analysis. This had been already explained in the text.

“In agreement with the AP2b2 NMR observations, no interaction was observed in the arrestin2 methyl and backbone NMR spectra upon addition of AP2b2 in the absence of phosphopeptide (Figure 3-figure supplement 1C, D). However, the significant line broadening of the arrestin2 resonances upon phosphopeptide addition (Figure 3-figure supplement 1E, F) precluded a meaningful assessment of the effect of the AP2b2 addition on arrestin2 in the presence of phosphopeptide”.

(3) I don't understand how methyl TROSY spectra of arrestin2 with phosphopeptide could look so broadened unless there are sample stability problems?

We thank the referee for this comment. We would like to clarify that in general a broadened spectrum beyond what is expected from the rotational correlation time does not necessarily correlate with sample stability problems. It is rather evidence of conformational intermediate exchange on the micro- to millisecond time scale.

The displayed ^1^H-^15^N spectra of apo arrestin2 already suffer from line broadening due to such intrinsic mobility of the protein. These spectra were recorded with acquisition times of 50 ms (^15^N) and 55 ms (^1^H) and resolution-enhanced by a 60° -shifted sine-bell filter for ^15^N and a 60° -shifted squared sine-bell filter for ^1^H, respectively, which leads to the observed resolution with still reasonable sensitivity. The ^1^H-^15^N resonances in Fig. 1b (arrestin2^1-393^) look particularly narrow. However, this region contains a large number of flexible residues. The full spectrum, e.g. Figure 1-figure supplement 2, shows the entire situation with a clear variation of linewidths and intensities. The linewidth variation becomes stronger when omitting the resolution enhancement filters.

The addition of the CCR5pp6 phosphopeptide does not change protein stability, which we assessed by measuring the melting temperature of arrestin2^1-418^ and arrestin2^1-418^•CCR5pp6 complex (Tm = 57°C in both cases). We believe that the explanation for the increased broadening of the arrestin2 resonances is that addition of the CCR5pp6, possibly due to the release of the arrestin2 strand b20, amplifies the mentioned intermediate timescale protein dynamics. This results in the disappearance of arrestin2 resonances.

We have now included the assessment of arrestin2^1-418^ and arrestin2^1-418^•CCR5pp6 stability in the manuscript:

“The observed line broadening of arrestin2 in the presence of phosphopeptide must be a result of increased protein motions and is not caused by a decrease in protein stability, since the melting temperature of arrestin2 in the absence and presence of phosphopeptide are identical (56.9 ± 0.1 °C)”.

(4) At one point the authors added excess fully phosphorylated CCR5 phosphopeptide (CCR5pp6). Does the phosphopeptide rescue resolution of arrestin2 (NH or methyl) to the point where interaction dynamics with clathrin (CLTC NTD) are now more evident on the arrestin2 surface?

Unfortunately, when we titrate arrestin2 with CCR5pp6 (please see Isaikina & Petrovic et. al, Mol. Cell, 2023 for more details), the arrestin2 resonances undergo fast-to-intermediate exchange upon binding. In the presence of phosphopeptide excess, very few resonances remain, the majority of which are in the disordered region, including resonances from the clathrin-binding loop. Due to the peak overlap, we could not unambiguously assign arrestin2 resonances in the bound state, which precluded our assessment of the arrestin2-clathrin interaction in the presence of phosphopeptide. We have made this now clearer in the paragraph ‘The arrestin2-clathrin interaction is independent of arrestin2 activation’

“Due to significant line broadening and peak overlap of the arrestin2 resonances upon phosphopeptide addition, the influence of arrestin activation on the clathrin interaction could not be detected on either backbone or methyl resonances “.

(5) Once phosphopeptide activates arrestin-2 and AP2 binds can phosphopeptide be exchanged off? In this case, would it be possible for the activated arrestin-2 AP2 complex to re-engage a new (phosphorylated) receptor?

This would be an interesting mechanism. In principle, this should be possible as long as the other (phosphorylated) receptor outcompetes the initial phosphopeptide with higher affinity towards the binding site. However, we do not have experiments to assess this process directly. Therefore, we rather wish not to further speculate.

(6) I'd be tempted to move the discussion of class A and class B GPCRs and their presumed differences to the intro and then motivate the paper with specific questions.

We appreciate the referee’s suggestion and had a similar idea previously. However, as we do not have data on other class-A or class-B receptors, we rather don’t want to motivate the entire manuscript by this question.

(7) Did the authors ever try SEC measurements of arrestin-2 + AP2beta2+CCR5pp6 with and without PIP2, and with and without clathrin CLTC NTD? The question becomes what the active complex is and how PIP2 modulates this cascade of complexation events in class B receptors.

We thank the referee for this question. Indeed, we tested whether PIP2 can stabilize the arrestin2•CCR5pp6•AP2 complex by SEC experiments. Unfortunately, the addition of PIP2 increased the formation of arrestin2 dimers and higher oligomers, presumably due to the presence of additional charges. The resolution of SEC experiments was not sufficient to distinguish arrestin2 in oligomeric form or in arrestin2•CCR5pp6•AP2 complex. We now mention this in the text:

“We also attempted to stabilize the arrestin2-AP2b2-phosphopetide complex through the addition of PIP2, which can stabilize arrestin complexes with the receptor (Janetzko et al., 2022). The addition of PIP2 increased the formation of arrestin2 dimers and higher oligomers, presumably due to the presence of additional charges. Unfortunately, the resolution of the SEC experiments was not sufficient to separate the arrestin2 oligomers from complexes with AP2b2”.

**Reviewer #3 (Public review):**
Summary:Overall, this is a well-done study, and the conclusions are largely supported by the data, which will be of interest to the field.Strengths:Strengths of this study include experiments with solution NMR that can resolve high-resolution interactions of the highly flexible C-terminal tail of arr2 with clathrin and AP2. Although mainly confirmatory in defining the arr2 CBL376LIELD380 as the clathrin binding site, the use of the NMR is of high interest (Fig. 1). The 15N-labeled CLTC-NTD experiment with arr2 titrations reveals a span from 39-108 that mediates an arr2 interaction, which corroborates previous crystal data, but does not reveal a second area in CLTC-NTD that in previous crystal structures was observed to interact with arr2.SEC and NMR data suggest that full-length arr2 (1-418) binding with 2-adaptin subunit of AP2 is enhanced in the presence of CCR5 phospho-peptides (Fig. 3). The pp6 peptide shows the highest degree of arr2 activation, and 2-adaptin binding, compared to less phosphorylated peptide or not phosphorylated at all. It is interesting that the arr2 interaction with CLTC NTD and pp6 cannot be detected using the SEC approach, further suggesting that clathrin binding is not dependent on arrestin activation. Overall, the data suggest that receptor activation promotes arrestin binding to AP2, not clathrin, suggesting theAP2 interaction is necessary for CCR5 endocytosis.To validate the solid biophysical data, the authors pursue validation experiments in a HeLa cell model by confocal microscopy. This requires transient transfection of tagged receptor (CCR5-Flag) and arr2 (arr2-YFP). CCR5 displays a "class B"-like behavior in that arr2 is rapidly recruited to the receptor at the plasma membrane upon agonist activation, which forms a stable complex that internalizes onto endosomes (Fig. 4). The data suggest that complex internalization is dependent on AP2 binding not clathrin (Fig. 5).The addition of the antagonist experiment/data adds rigor to the study.Overall, this is a solid study that will be of interest to the field.

We thank the referee for the careful and encouraging evaluation of our work. We appreciate the recognition of the solidity of our data and the support for our conclusions regarding the distinct roles of AP2 and clathrin in arrestin-mediated receptor internalization.

**Recommendations for the authors:**

**Reviewer #1 (Recommendations for the authors):**
I believe that the authors have made efforts to improve the accessibility to a broader audience. In a few cases, I believe that the authors response either did not truly address the concern or made the problem worse. I am grouping these as 'very strong opinions' and 'sticking point'.Very strong opinion 1:While data presentation is somewhat at the authors discretion, there were several figures where the presentation did not make the work approachable, including microscopy insets and NMR spectra. A suggestion to 'pinch and zoom' does not really address this. For the overlapping NMR spectra in supporting Figure 1, I actually -can- see this on zooming, but I did not recognize this on first pass because the colors are almost identical for the two spectra. This is an easy fix. Changing the presentation by coloring these distinctly would alleviate this. The Supplemental figure to Fig. 2 looks strange with pinch and zoom. But at the end of the day, data presentation where the reader is to infer that they must zoom in is not very approachable and may prevent readers from being able to independently assess the data. In this case, there doesn't seem to be a strong rationale to not make these panels easier to see at 100% size.

We appreciate the reviewer’s thoughtful comments regarding figure accessibility and agree that data presentation should be clear and interpretable without requiring readers to zoom in extensively. However, we must note that the presentation of the microscopy data follows standard practices in the field and that the panels already include zoomed-in regions, which enable easier access to key results and observations.

Regarding the NMR data, we have revised Figure 1—figure supplement 2 and Figure 2— figure supplement 1 to match the presentation style of Figure 3—figure supplement 1, which the reviewer apparently found more accessible. We also made the colors of the spectra more vibrant, as the referee suggested. We would like to emphasize that it is absolutely necessary to display the full NMR spectra in order to allow independent assessment of signal assignment, data quality, and overall protein behavior. Zoomed regions of the relevant details are provided in the main figures.

Very strong opinion 2:The author's response to lack of individual data points and error bars is that this is an n=1 experiment. I do not believe this meets the minimum standard for best practices in the field.

We respectfully disagree with the reviewer’s assessment. The Figure already displays individual data points, as shown already in the initial submission. Performing NMR titrations with isotopically labeled protein samples is inherently resource-intensive, and single-sample (n = 1) experiments are widely accepted and routinely reported in the field. Numerous studies have used the same approach, including Rosenzweig et al., Science (2013); Nikolaev et al., Nat. Methods (2019); and Hobbs et al., J. Biomol. NMR (2022), as well as our own recent work (Isaikina & Petrovic et al., Mol. Cell, 2023). These studies demonstrate that such NMR-based affinity measurements, even when performed on a single sample, are highly reproducible, precise, and consistent with orthogonal evidence and across different sample conditions.

Sticking point:Figure 1A - the alphaFold model of arrestin2L depicts the disordered loops as ordered. The depiction is misleading at best, and inaccurate in truth. To use an analogy, what the authors depict is equivalent to publishing an LLM hallucination in the text. Unlike LLMs, alphaFold will actually flag its hallucination with the confidence (pLDDT) in the output. Both for LLMs and for alphaFold, we are spending much time teaching our students in class how to use computation appropriately - both to improve efficiency but also to ensure accuracy by removing hallucinations.The original review indicated that confidences needed to be shown and that this needed to be depicted in a way that clarifies that this is NOT a structural state of the loops. The newly added description ("The model was used to visualize the clathrin-binding loop and the 344-loop of the arrestin2 Cdomain, which are not detected in the available crystal structures...) worsens the concern because it even more strongly implies that a 0 confidence computational output is a likely structural state. It also indicates that these regions were 'not detected' in crystal structures. These regions of arrestin are intrinsically disordered. AlphaFold (by it's nature) must put out something in terms of coordinates, even if the pLDDT suggests that the region cannot be predicted or is not in a stable position, which is the case here. In crystal structures, these regions are not associated with interpretable electron density, meaning that coordinates are omitted in these regions because adding them would imply that under the conditions used, the protein adopts a low energy structural state in this region. This region is instead intrinsically disordered.A good description of why showing disordered loops in a defined position is incorrect and how to instead depict disorder correctly is in Brotzakis et al. Nat communications 16, 1632 (2025) "AlphaFold prediction of structural ensembles of disordered proteins", where figures 3A, 4A, and 5A show one AlphaFold prediction colored by confidence and 3B, 4B and 5B are more accurate depictions of the structural ensemble.Coming back to the original comment "The AlphaFold model could benefit from a more transparent discussion of prediction confidence and caveats. The younger crowd (part of the presumed intended readership) tends to be more certain that computational output is 'true'...." Right now, the authors are still showing in Fig 1A a depiction of arrestin with models for the loops that are untrue. They now added text indicating that these loops are visualized in an AlphaFold prediction and 'true' but 'not detected in crystal structures'. There is no indication in the text that these are intrinsically disordered. The lack of showing the pLDDT confidence and the lack of any indication that these are disordered regions is simply incorrect.

We appreciate the concern of the reviewer towards AlphaFold models. As NMR spectroscopists we are highly aware of intrinsic biomolecular motions. However, our AlphaFold2 model is used as a graphical representation to display the interaction sites of loops; it is not intended to depict the loops as fixed structural states. The flexibility of the loops had been clearly described in the main text before:

“Arrestin2 consists of two consecutive (N- and C-terminal) β-sandwich domains (Figure 1A), followed by the disordered clathrin-binding loop (CBL, residues 353–386), strand b20 (residues 386–390), and a disordered C-terminal tail after residue 393”.

and

“Figure 1B depicts part of a 1H-15N TROSY spectrum (full spectrum in Figure 1-figure supplement 2A) of the truncated 15N-labeled arrestin2 construct arrestin21-393 (residues 1393), which encompasses the C-terminal strand β20, but lacks the disordered C-terminal tail. Due to intrinsic microsecond dynamics, the assignment of the arrestin21-393 1H-15N resonances by triple resonance methods is largely incomplete, but 16 residues (residues 367381, 385-386) within the mobile CBL could be assigned. This region of arrestin is typically not visible in either crystal or cryo-EM structures due to its high flexibility”.

as well as in the legend to Figure 1:

“The model was used to visualize the clathrin-binding loop and the 344-loop of the arrestin2 C-domain, which are not detected in the available crystal structures of apo arrestin2 [bovine: PDB 1G4M (Han et al., 2001), human: PDB 8AS4 (Isaikina et al., 2023)]. In the other structured regions, the model is virtually identical to the crystal structures”.

We have now further added a label ‘AlphaFold2 model’ to Figure 1A and amended the respective Figure legend to

“The model was used to visualize the clathrin-binding loop and the 344-loop of the arrestin2 C-domain, which are not detected in the available crystal structures of apo arrestin2 [bovine: PDB 1G4M (Han et al., 2001), human: PDB 8AS4 (Isaikina et al., 2023)] due to flexibility. In the other structured regions, the model is virtually identical to the crystal structures”.

**Reviewer #2 (Recommendations for the authors):**
I appreciated the response by the authors to all of my questions. I have no further comments

We thank the referee for the raised questions, which we believe have improved the quality of the manuscript.